# EUBREWNET RBCC-E Huelva 2015 Ozone Brewer Intercomparison

Alberto Redondas[1,2], Virgilio Carreño[1,2], Sergio F. León-Luis[1,3], Bentorey Hernández-Cruz[2,3], Javier López-Solano[1,2,3], Juan J. Rodriguez-Franco[1,3], José M. Vilaplana[4], Julian Gröbner[5], John Rimmer[6], Alkiviadis F. Bais[7], Volodya Savastiouk[8], Juan R. Moreta[9], Lamine Boulkelia[10], Nis Jepsen[11], Keith M. Wilson[12], Vadim Shirotov[13], and Tomi Karppinen[14]

[1]Izaña Atmospheric Research Center, Agencia Estatal de Meteorología, Tenerife, Spain
[2]Departamento de Ingeniería Industrial, Universidad de La Laguna, Tenerife, Spain
[3]Regional Brewer Calibration Center for Europe, Izaña Atmospheric Research Center, Tenerife, Spain
[4]National Institute for Aerospace Technology - INTA, Atmospheric Observatory "El Arenosillo", Huelva, Spain
[5]Physikalisch-Meteorologisches Observatorium Davos/World Radiation Center, Davos, Switzerland
[6]Manchester University, Manchester, United Kingdom
[7]Laboratory of Atmospheric Physics, Aristotle University of Thessaloniki, Thessaloniki, Greece
[8] International Ozone Services, Toronto, Canada
[9]Agencia Estatal de Meteorología, Madrid, Spain
[10]National Meteorological Office, Algeria
[11]Danish Meteorological Institute, Copenhagen, Denmark
[12]Kipp & Zonen, Delft, The Netherlands
[13]Scientific and Production Association "Typhoon", Obninsk, Russia
[14]Finnish Meteorological Institute, Sodankyla, Finland

*Correspondence to:* Alberto Redondas (aredondasm@aemet.es)

**Abstract.**

From May 25th to June 5th 2015, the 10th Regional intercomparison campaign of the Regional Brewer Calibration Center-Europe (RBCC-E) was held at El Arenosillo atmospheric sounding station of the Instituto Nacional de Técnica Aeroespacial (INTA). This campaign was jointly conducted by COST Action ES1207 EUBREWNET and the Area of Instrumentation and Atmospheric Research of INTA. Twenty one Brewers, 11 single and 10 double monochromator instruments from eleven countries participated and were calibrated for total column ozone (TOC) and solar UV irradiance. In this 2015 campaign we have introduced a formal approach to the characterization of the internal instrumental stray light, the filter non-linearity and the algorithm for correcting for its effects on the TOC calculations. This work shows a general overview of the ozone comparison and the evaluation of the correction of the spectral stray light effect for the single-monochromator Brewer spectrophotometer, derived from the comparison with a reference double-monochromator Brewer instrument. At the beginning of the campaign, 16 out of the 21 participating Brewer instruments agreed within better than $\pm$ 1%, and 10 instruments agreed within better than $\pm$ 0.5% considering data with Ozone Slant Column between 100 and 900 DU, which doesn't require instrumental stray light correction.

# 1 Introduction

The fully automated Brewer Spectrophotometer (Brewer, 1973; Kerr et al., 1985; Kerr, 2010) is together with the Dobson ozone spectrophotometer, the backbone of the World Meteorological Organization (WMO) ozone observation network providing high quality Total Ozone Column (TOC) data for more than 30 years and is now deployed at 200 ground based TOC monitoring stations worldwide. It is also capable of measurements of ozone vertical profiles (Umkehr method), spectral UV radiation and aerosol optical depth in the UV (AOD-UV), as well as columns of other trace constituents such as sulphur dioxide and nitrogen dioxide.

In November 2003 the WMO/GAW Regional Brewer Calibration Center for Europe (RBCC-E) was established at the Izaña Atmospheric Observatory (IZO) of the Agencia Estatal de Meteorología (AEMET) in Tenerife (Canary Islands, Spain). The RBCC-E consists of calibration laboratory facilities and reference-maintenance equipment mainly composed of three Brewer spectrophotometers, the denoted IZO Triad. This includes a Regional Primary Reference (Brewer #157), a Regional Secondary Reference (Brewer #183), and a Regional travelling Reference (Brewer #185) which can be transported for calibration campaigns outside IZO. Initially, the RBCC-E transferred the calibration from the World Reference Triad in Toronto. However, due to uncertainties on the future maintenance of the World Triad, in 2011, the WMO scientific advisory group (WMO-SAG) authorized the RBCC-E to transfer its own calibration obtained by the Langley method.

RBCC-E regular intercomparisons are held annually, alternating between Arosa in Switzerland, and the El Arenosillo sounding station of the Instituto Nacional de Técnica Aeroespacial (INTA) at Huelva in the south of Spain. Since 2005, a total of 130 Brewer ozone spectrophotometer calibrations have been performed in these campaigns (see the campaign reports at the RBCC-E website, http://rbcce.aemet.es, and the GAW reports of the VII (Redondas et al., 2015), VIII (Redondas and Rodriguez-Franco, 2016), and IX (Redondas and Rodriguez-Franco, 2015b) intercomparison campaigns). In addition to the regular intercomparisons, the RBCC-E performs two types of campaigns supported by the ESA (European Space Agency) Validation projects: the NORDIC campaigns, with the objective to study the ozone measurements at high latitudes, and the Absolute calibration campaigns performed at IZO with the participation of Brewer and Dobson reference instruments. Figure 1 shows the number of Brewer instruments calibrated at these campaigns since 2003.

The aim of COST Action 1207 "EUBREWNET" is to establish a coherent network of European stations equipped with Brewer spectrophotometers for the monitoring of total ozone column (TOC), spectral UV radiation, and aerosol optical depth in the UV spectral range (AOD-UV), ensuring sustainable operation in the long-term (Rimmer et al., 2018). Among the primary aims of EUBREWNET is to harmonise operations and develop approaches, practices and protocols to achieve consistency in quality control, quality assurance and coordinated operations, as well as to eliminate duplication of efforts at individual stations. It also aims at establishing knowledge exchange and training, and at opening up a route to link with international agencies and other networks globally.

It was proposed by Fioletov et al. (2008) that the main problem of the Brewer network was the lack of QA/QC, the non-uniformity of the data processing, and the lack of reprocessing of the observations after the calibration. This problem is also aggravated because there are more than 100 agencies providing Brewer data with different calibration practices, operational

procedures and data processing, including many of these stations within Europe. The intercomparisons are a basic tool to achieve EUBREWNET objectives, in particular during RBCC-E calibration campaigns the instruments perform a common measurement schedule, the observations are uniformly processed and can be used to organise operator training on applying uniform operational procedures. During these campaigns we are able to address some important issues which affect the Brewer

performance, these include instrumental parameters and the calibration methodology (Redondas and Rodriguez-Franco, 2012)

Before the establishment of the RBCC-E, the Brewer spectrophotometer calibrations were referenced to the Brewer World Calibration Center hosted by Environmental Canada (EC). However, most of the Brewer instrument were, and still are, calibrates by private companies, in the main by International Ozone Servicies (IOS) and to a lesser extent by Kipp and Zonen (Staehelin, 2010). The RBCC-E calibration adapts the methodologies and tools developed by EC and IOS, but also investigate

and improves particular issues. The focus in the first campaigns were related to the instrument characterisation and the ozone absorption calculation (Redondas and Rodriguez-Franco, 2012) whereas in this campaign the focus was on the stray light correction and the investigation of the error due to non-linear filter attenuation.

The EUBREWNET RBCC-E Huelva 2015 Ozone Brewer Intercomparison campaign had two main objectives: to establish the current status of the network with the participation of about the half of the participating Brewers in the network and to test

the improvements introduced during the COST action on the total ozone processing (the stray light correction on single Brewer and the attenuation filter corrections). As well as ozone calibration, solar UV irradiance calibration was also performed by the travelling reference standard QASUME instrument of the World Calibration Center for UV (WCC-UV) and is described by the companion paper by (Lakkala et al., 2017), also it is remarkable that this campaign provided the AOD-UV (López-Solano et al., 2017).

The present work is organised as follows. The Brewer algorithm for TOC and the calibration methodology implemented during the RBCC-E campaigns is described in Sect 2, with focus on the improvements introduced by the stray light and the filter corrections. In Sect. 3 we describe and present results of intercomparisons. In the blind days section we check the stability of the TOC for the approximately 2-year period between calibrations and the final days subsection shows the agreement of the instrument at the end of the campaign. In Sect. 4 we discuss the results and the implications of the correction introduced by the

EUBREWNET processing and provide some closing remarks.

## 2 The calibration of the Brewer spectrophotometer

A brief description of the instrument and its operating principles in relation to the calibration is described in this section. A more detail description can be found in (Kerr, 2010; Savastiouk, 2006) .

The Brewer instrument measures the intensity of direct sunlight at six wavelengths ($\lambda$) in the UV (303.2, 306.3, 310.1,

313.5, 316.8, and 320.1 nm) each covering a bandwidth of 0.5 nm (resolution power $\lambda/\delta\lambda$ of approximately 600). The spectral measurement is achieved by a holographic grating in combination with a slit mask which selects the channel to be analysed by a photomultiplier. The longest four wavelengths are used for the ozone calculation. The basic ozone measurement is the direct sun measurement. The sunlight enters the instrument through an inclined quartz window after which a right-angle prism directs

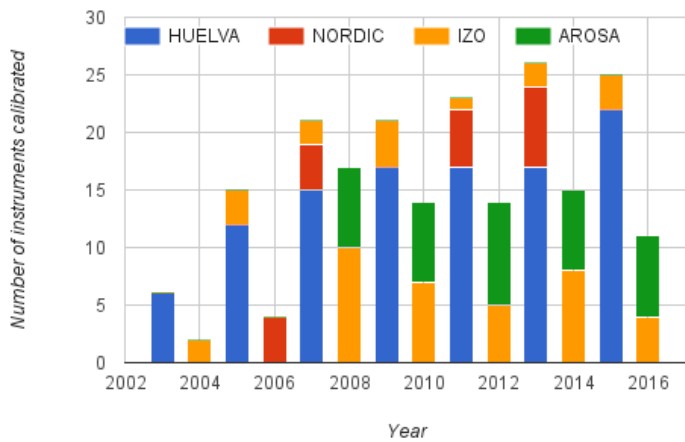

**Figure 1.** Brewer instruments calibrated since 2003 by the RBCC-E in regular campaigns (at Huelva and Arosa), Nordic intercomparisons, and the Absolute calibrations performed at the Izaña Observatory.

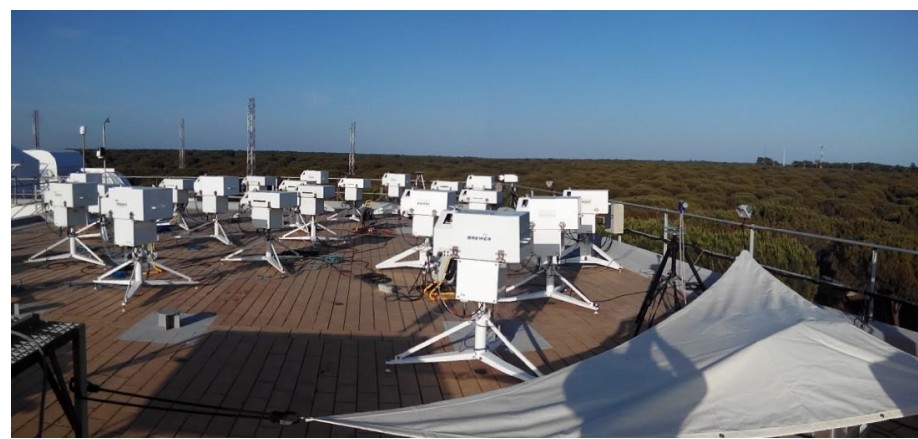

**Figure 2.** Panoramic view of the 21 Brewer spectrophotometers on the terrace of the El Arenosillo sounding station, Huelva, coming from Canada (1), Netherlands (2), United Kingdom (3), Switzerland (1), Finland (1), Greece (1), Denmark (2), Russia (1), Algeria (1) and Spain(7).

the incoming light to the optical axis of the instrument. The light subsequently passes through the fore-optics, which consist of a set of lenses to adequately focus the beam, an iris diaphragm, and two filter wheels. A ground quartz diffuser is located on the first filter wheel. The second filter wheel consists of a set of five neutral density filter attenuators which guarantee that the detector is working in its linear regime. After passing through the filter wheels, radiation is then focused onto the entrance slit of the monochromator. Presently, there are three commercially available installed Brewer models. The oldest model MKII

is a single monochromator with a solar-blind filter, a $NiSO_4$ element sandwiched between two UG-11 glass filters, located between the exit slit and the PMT to block radiation at wavelengths longer than about 325 nm. The model MKIV is capable of measuring $NO_2$, this instrument works with its grating in the third diffraction order for the ozone retrievals and the second diffraction order in $NO_2$ retrievals. However, both instruments are affected by stray light, mainly, because of using a single monochromator. In contrast, the the model MKIII is a double monochromator which provides enough stray light rejection to work in the first diffraction order for UV and ozone measurements.

Based on the Lambert-Beer law, the total ozone column in the Brewer algorithm can be expressed as (Kerr, 2010)

$$X = \frac{ETC - F}{\alpha\mu} \tag{1}$$

where $F$ is a linear combination of the logarithm of the measured spectral direct irradiances (also called double ratios) corrected for Rayleigh molecular scattering, $\alpha$ is the ozone differential absorption coefficient, $\mu$ is the ozone air mass factor, and $ETC$ is the extra-terrestrial constant. The $F$, $\alpha$ and $ETC$ parameters are weighted functions at the operational wavelengths:

$$F = \sum_i^4 w_i F_i - \frac{p}{p_0}\beta_i\mu \tag{2}$$

$$\alpha = \sum_i^4 w_i\alpha_i \tag{3}$$

$$ETC = \sum_i^4 w_i F_{0i} \tag{4}$$

where $\beta_i$ are the Rayleigh coefficients, $p$ is the climatological pressure at the measurement site, $p_0$ is the pressure at sea level, and $F_i$ and $F_{0i}$ are the individual measured and extra-terrestrial irradiances at each wavelength respectively. The four longer wavelengths (310.1, 313.5, 316.8 and 320.1 nm are used on the ozone calculation with the respective weights of $w = [1, -0.5, -2.2, 1.7]$. These wavelengths have been selected near stationary points in the ozone absorption spectrum and are thus optimised to minimise the influence of small wavelengths shift (i.e. $\delta F//\delta\lambda = 0$). The weightings were determined to suppress the influence of $SO_2$ and aerosol. Moreover as $\lambda_i$ and $w_i$ satisfy the conditions defined by Eqs. (5) and (6), the measurement not dependent on wavelength-independent parameters such as the absolute calibration. Also, it largely eliminates absorption processes which depend, to first approximation, linearly on the wavelength such as the contribution from aerosols (Kerr, 2010).

$$\sum_i^4 w_i = 0 \tag{5}$$

$$\sum_{i}^{4} w_i \lambda_i \approx 0 \qquad (6)$$

The Brewer retrieval of the TOC requires the knowledge of some instrument characteristics which are determined by calibration experiments during intercomparison campaigns (Redondas et al., 2015; Redondas and Rodriguez-Franco, 2015a, b, see e.g. the GAW reports of the Seventh, Eighth, and Ninth Intercomparison Campaigns of the RBCC-E). The instrumental calibration includes all the parameters that affect the counts measured by the spectrometer ($F_i$), in particular the dead time correction (Fountoulakis et al., 2016), temperature coefficients (Berjón et al., 2017), and filter attenuations. The wavelength calibration determines the ozone absorption and Rayleigh scattering coefficients. The exact wavelengths measured by each Brewer spectrophotometer are slightly different from instrument to instrument. The so-called "dispersion test" is thus used to determine the exact wavelengths of each instrument and its slit, or instrumental, function (Gröbner et al., 1998; Redondas et al., 2014a). An extra-terrestrial (calibration) constant is determined by the Langley method (Redondas et al., 2014b; León-Luis et al., 2018) , in the case of reference brewers , or by comparison with a reference instrument in the case of the other Brewer of the Network.

It is important to note that TOC It is then finally determined using ratios of measurements so there is no transfer of the radiometric scale. During the campaigns the transfer of the calibration to a network instrument is achieved by operating side by side with the reference Brewer. Once we have collected enough near-simultaneous direct sun ozone measurements, we calculate the new extra-terrestrial constant after imposing the condition that the measured ozone will be the same for simultaneous measurements for both instruments. In terms of Eq. 1, this leads to the following condition:

$$ETC_j = F_j + X_j^{reference} \alpha \mu_j \qquad (7)$$

For a correctly characterised network instrument, the determined ETC values show a Gaussian distribution and the mean value is used as the instrument's extra-terrestrial constant. One exception to this rule is the single monochromator Brewer models (MK-II and MK-IV) which are affected by stray light (Karppinen et al., 2015). In this case, the ETC distribution shows a tail at the lower ETC values for high Ozone Slant Column (OSC, the product of the total ozone content by the airmass). As we discuss in detail on the next section for this type of Brewer, only the stray-light-free region is used to determine the ETC, which generally ranges from 300 to 800 DU in the OSC, depending on the instrument.

The network Brewers were calibrated using the one parameter ETC transfer method: i.e., the ozone differential absorption coefficient was derived from the calculations of wavelength calibration, the so-called "dispersion test" (Redondas et al., 2018-01-31), applied to the spectroscopic set of the ozone cross section and, only the ozone ETC constant was transferred from the reference instrument. The so-called "two parameters calibration method" (Staehelin et al., 2003), where both the ozone absorption coefficient and the ETC are calculated from the reference, is also obtained and used as a quality indicator.

The calibration is an iterative process because changes during the instrumental and/or wavelength calibration will affect the final ETC. Some instrumental characteristics which have been improperly accounted such as the non-linearity of the filters

and the dead-time mismatch (Rodriguez-Franco et al., 2014), are revealed by the comparison with the reference during the ETC transfer. A change in the instrumental constants then requires a full reprocessing of the calibration. For this reason the calibration campaigns are scheduled in three different period:

1. **Blind days:** the first days of the campaign are dedicated to determine the current status of the instrument by comparison with the reference instrument. During this period modifications of the instrument are not allowed.

2. **Characterisation:** after the determination of how the instrument is measuring, the next days are dedicated to characterise the instrument and perform the necessary adjustments and maintenance. The instrumental and wavelength calibration must be finished at the end of this period.

3. **Final days:** the period where the ETC transfer is performed, when the instrument is fully characterised and stable.

The changes in the spectral sensitivity of the instruments are tracked between calibration transfers using the measurements of the internal quartz halogen lamp, the so-called Standard Lamp (SL) test. A value, corresponding to a fictitious column density and often called R6, R5 or F-ratio (depending on whether the ozone, sulphur dioxide or nitrogen dioxide processing algorithm is applied), is obtained after processing. Slow variations of the SL test results with time may be representative of a change in the relative spectral (but not absolute, because of the way the retrieval algorithm works) sensitivity of the Brewer and may be even used to correct the final value of the solar measurements.This test is performed routinely to track the spectral response of the instrument and, therefore, the ozone calibration. A reference value for the SL, the so-called R6 ratio, is provided as part of the calibration of the instrument. The ozone is routinely corrected assuming that deviations of the R6 value from the reference value are the same as the changes in the ETC Extraterrestrial constant. This is then described by the Standard Lamp correction:

$$ETC_{new} = ETC_{old} - (SL_{ref} - SL_{measured}) \tag{8}$$

## 2.1 Stray light

There are two major sources of stray light on the Brewer: the "sky scattered" and the "out of band stray light, both of them produce an underestimation on the ozone calculation. The "sky scattered" stray light is due to the instrument having a field of view (FOV) larger than the apparent angle subtended by the sun in the sky, approximately 0.5º whereas the instrument FOV is about 1.5º, the measurement will thus include a "sky scattered" component (Josefsson, 2003, 2012).

The second source of stray light is due to the finite dimensions of the exit slits and imperfections on the gratings or other optical components so that the intensity measured contains radiation from other unwanted wavelengths. The higher level of stray light in single monochromators increase the error of the measured UV radiation compared with the double-monochromators. This effect is largest at short UV wavelengths where the spectral irradiance level is low compared with the level of potential stray light. It is known to affect the Brewer spectral UV measurement (Kerr and McElroy, 1993; Bais et al., 1996) but only relatively recently has the effect on the ozone calculation been studied (Bojkov et al., 2008; Kiedron et al., 2008; Evans et al.,

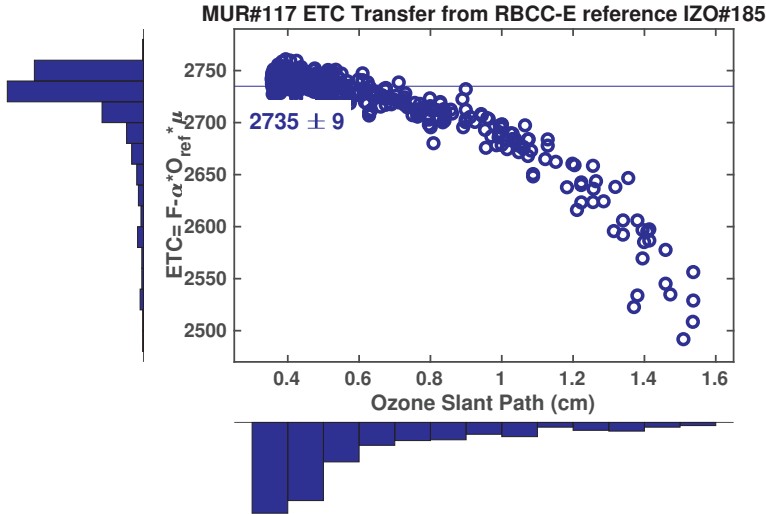

**Figure 3.** Distribution of individual ETC values determined by simultaneous measurements. In the horizontal axis, the ozone slant column (OSC). For this particular Brewer , the effect of the stray light is strong and clearly shown at values above $0.6$ cm.

2009; Petropavlovskikh et al., 2011). The effect on the ozone calculation is enhanced when the short wavelength proportion is decreased due to a high slant path or high ozone content. The air mass in combination with the total ozone amount gives a measure of the absorption of the UV photons and is the key parameter for the stray light. There have been several attempts to model the stray light (e.g. Karppinen et al. (2015), Pulli et al. (2018) and the references within) but this requires the use of
5  models or characterisation of the instruments that is not always available. Here we propose an empirical method based on the comparison of a double Brewer.

    The stray light effect can be corrected if the calibration is performed against a double monochromator instrument, assuming that it can be characterised following a power law of the ozone slant column:

$$F = F_m + k(X\mu)^s \tag{9}$$

10  where $F$ are the true weighted ratios (Eq 2) , $F_m$, the measured ones, and the k parameter are negative. This is equivalent to correcting the extraterrestrial constant.

$$ETC_i = ETC_0 + k(X\mu)^s \tag{10}$$

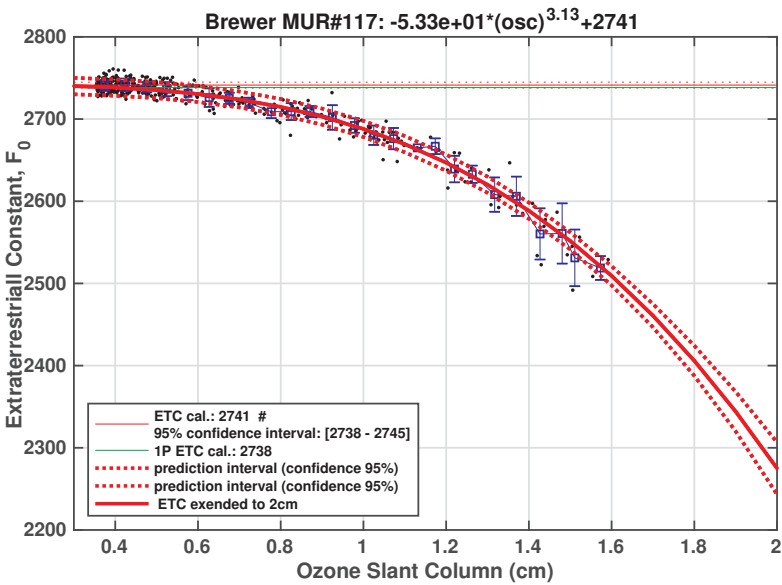

**Figure 4.** The stray light parameters $k$ and $s$ are determined by a nonlinear fit using the ETC determined from the stray-light-free region as first guess parameters. The red horizontal line indicate the ETC constant retrieved from the fit, and the green one, the initial guess. The upper and lower 95% prediction bounds are also displayed in red dot lines.

where $ETC_0$ is the ETC for the OSC region free of stray light, and $k$ and $s$ are retrieved from the reference comparison (Figure 4). These parameters, determined in several campaigns, have been found to be stable and independent of the ozone calibration.

As the counts ($F$) from the single Brewer instrument are affected by stray light, the ozone is calculated using an iterative process:

$$X_{i+1} = X_i + \frac{k(X_i\mu)^s}{\alpha\mu} \tag{11}$$

This empirical method was tested during the NORDIC campaigns and the counterpart campaigns at Izaña with the MKII Brewer #037 operated by the Finish Meteorological Institute at Sodankyla (Finland) since 1988 (Karppinen et al., 2016). This instrument was calibrated four times at Izana in 2009, 2011, and 2015 and at Sodankyla in 2011 (Roozendael et al., 2013b, 2014). From these measurement campaigns we found the stray light correction obtained during the first campaign can be applied to the subsequent campaigns, obtaining a good agreement better than 0.5% on the 300-1800 OSC range. This is confirmed with the agreement of the determination of the stray light parameters k and s obtained during different campaigns (Table 3 of (Roozendael et al., 2014)) at different locations with quite different sky conditions and even with changes on spectral response of the instrument. A similar experiment was done at Huelva with a good agreement using the previous campaign determination of the stray light parameters.

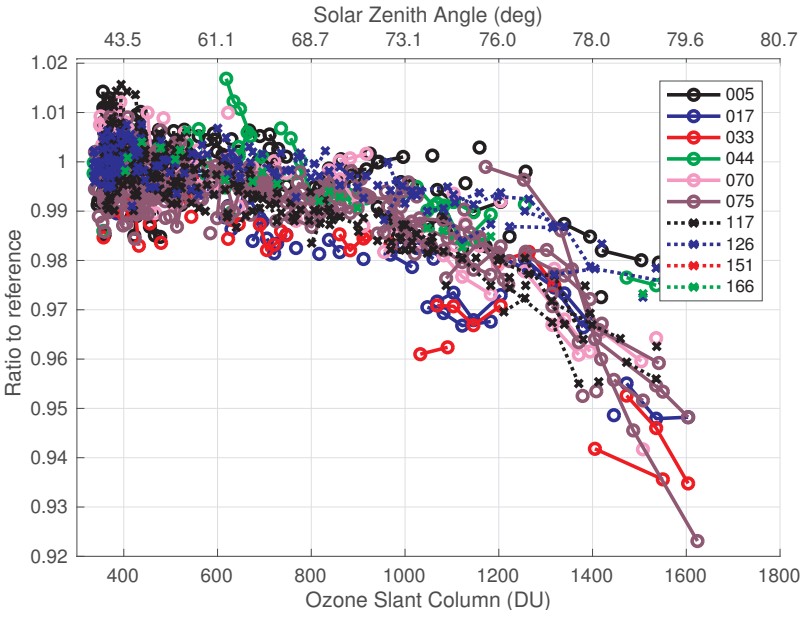

**Figure 5.** Percentage ozone differences with respect to the reference of the participating Single Brewers. On the upper x-axis the approximate solar zenith angle is indicated assuming TOC of 300 DU

Figure 5 shows the ratio of the TOC calculated by single Brewer participating in the campaign respect to reference instrument, where we can see that the stray light-free region which we use for calibration ends for some instruments at 600 DU and is almost evident at 1000DU. The error due to the extrapolation can be estimated from the fit, these error bounds will depend on every instrument and strongly on the extrapolation (Fig 4). During the Nordic campaign the error bounds for Brewer
#037 at 2000 DU increases from +/- 0.25%, if we use observations up to 1600 DU, to 1.7% if we use only observations up to 1200 DU. During this campaign with observations up to 1600 DU, the error bounds at 2000 DU are lower than 2% for all the instruments. This results are consistent with the results obtained from the model of Karpinen that shows an error of 1.29 % on 1900-2000 DU interval ( Table 2 of Karppinen et al. (2015)). To help the determination of these parameters the measurement schedule is carefully defined to maximize the observations at high OSC, where around 30% of the simultaneous observations
are performed with OSC> 600 and 15% are obtained with OSC >900 DU.

Usually just one iteration is needed for the atmospheric conditions at the intercomparisons carried out at El Arenosillo, with OSC values up to 1500 DU. For OSC measurements in the 1500–2000 DU range, two iterations are enough to correct the ozone (Figure 7). These stray light corrections are now implemented in the standard procedure of EUBREWNET.

The effect of the stray light has been studied in the Sodankyla total ozone series, one of the longest record in the Artic
(Roozendael et al., 2014). In this location even at noon the ozone slant column can be quite large. When the model from Karppinen et al. (2015) is applied to the series the effect on the monthly means is significant in spring but with a marginal

effect on the trends. The comparison between the empirical correction and the model shows a good agreement (0.1 %) in most of the operative range but the model underestimate above 1600 DU reaching 1% at 2000 DU.

## 2.2 Correction of filter non-linearity

The Brewer spectrophotometer uses neutral density filters in order to optimise the intensity of the light which reaches the photomultiplier. The ozone is calculated using ratios so if the filters are neutral they have no effect on the ozone calculation (Eq. 5). In addition the weighing coefficients also verify (Eq. 6) so a linear attenuation with wavelength also does not affect the ozone calculation. In a real instrument however, some of the filters are not neutral, the ozone wavelengths vary from instrument to instrument, and the second condition is only an approximation. The effect on the calculated ozone will depends on the filter used (Savastiouk, 2006; Redondas et al., 2011). This dependence is clear when the instrument is compared with a reference and the comparison is seen to depend on the filter (Fig 6). Differences up to 20 ETC units (up to 4% in ozone) have been observed during the campaign.

This error can be corrected if we know the spectral dependence of the filter $AF_i$ and introduce a correction dependent of the filter used (f#)

$$F(f\#) = F_m + \sum_{i=1}^{4} w_i AF(f\#)_i$$

where $F$ are the true weighted ratios (Eq 2) and $F_m$, the measured ones.

The effect on ozone is only important at low air masses, where the ozone calculation is more dependent of the ETC, which implies high solar signal and in consequence high attenuation filters used.

Several methods have been developed to determine the wavelength dependence of the filter, using the internal lamp, the sun as source. However, as it affects high attenuation filters, there is not enough signal with the lamp and the non-linearity is difficult to determine with precision. Frequently the results are not significant with the number of tests that can be performed during a campaign, and the comparison with a well characterised instrument is the preferred method during the intercomparsions. Also we can determine the correction directly by examining the record of the instrument and looking for the simultaneous measurements performed with consecutive filters or determine the ETC constant for every filter.

## 3 Intercomparison Results

### 3.1 The X RBCC-E campaign

From May 25$^{th}$ to June 5$^{th}$ 2015, 21 Brewer spectrophotometers from 11 countries (see Table 1) took part in the X RBCC-E campaign held at the El Arenosillo atmospheric sounding station (Huelva, Spain). Besides the ozone calibration, a solar UV irradiance calibration was performed by the travelling reference standard QASUME (Hülsen et al., 2016) instrument of the World Calibration Center for UV (WCC-UV). The X RBCC-E campaign was the result of the collaboration between COST Action

1207 "EUBREWNET" (http://www.eubrewenet.org/cost1207), and the Area of Instrumentation and Atmospheric Research of INTA. (Redondas et al., 2016).

El Arenosillo Atmospheric Sounding Station, which belongs to the National Institute for Aerospace Technology (INTA), is located on the Atlantic Coast, in the province of Huelva of the region of Andalusia, in south-west Spain. Its surroundings
correspond to the "Doñana" National Park, which guarantees its natural environment. Moreover, the climate at "El Arenosillo" is characterised by very frequent sunny conditions, around 280 clear sky days per year, being a really suitable site for intercomparison campaigns. The observatory has a big terrace with completely open horizon Fig(2). The surroundings of the station consist of pine trees which provide a uniform albedo spatially and temporally throughout the whole year. This constant behaviour of the albedo allows the comparability of results obtained in different seasons and years.

We collected during the campaign $\approx 650$ direct sun ozone measurements with the reference instrument, most of them ($\approx$ 65%) within the 300-600 DU and 18% on the 600-900 DU and 17% 900-1700 DU ozone slant path range, in larger airmass. The mean number of near-simultaneous ozone measurements between the Brewers and the reference instrument was 350. Total ozone content values at El Arenosillo station during the intercomparison ranged between 320 to 380 DU. This campaign was characterised by high internal temperatures, with an average of 32°C and a standard deviation of 5°C, which is 10°C above the
spectrometer normal operating temperatures (Berjón et al., 2017).

Briefly the Brewer network is calibrated with two main parameters: the ETC and the effective ozone absorption coefficient. The comparison with a reference instrument is used to establish the ETC but the effective ozone absorption coefficient can be derived also from the reference (two parameters method) as was done in the past (Staehelin et al., 2003) or directly from wavelength calibration as is performed at present (one parameter method) (Fioletov et al., 2005). Historically the two parameter
method was used until around year 2000 after that the one parameter method has been adopted in the Brewer network. Although both methods give the same results in the 300-800 DU range (stray light free range), the two parameters calibration gives uniform results, smoothing the instrumental differences and reduces the stray light error on single Brewer (Bojkov et al., 2008). On the other hand, the one parameter calibration is more robust, it does not depend on reference wavelength calibration, and it highlights instrumental differences. An error in the ETC value yields an error that depends on the solar zenith angle,
while an error in effective ozone absorption coefficient introduces a relative bias. The transition in the calibration methodology around the year 2000 can explain the change of the seasonal difference between ground base Brewer and satellites observed by Fioletov et al. (2008). During the RBCC-E campaigns we can show that the two point calibration can mask instrumental issues which are air-mass dependent. Moreover, both calibration methods give the same results on well characterised instruments and the difference between the calibration constants can be used as an indicator of the quality of the instrument calibration
(Redondas and Rodriguez-Franco, 2012; Roozendael et al., 2014).

## 3.2    Reference Calibration

The RBCC-E triad is regularly calibrated, performing the instrumental characterisation and wavelength calibration monthly. The three instruments are independently calibrated by the Langley plot method following the procedure described in Redondas et al. (2014b) and León-Luis et al. (2018). Before and after the intercomparison campaigns, the travelling instrument is com-

pared with the two static instruments to verify that the calibration has not changed during transport (Figures 8 ), (Redondas et al., 2015; Redondas and Rodriguez-Franco, 2015a, b; León-Luis et al., 2018).

The campaign is a good opportunity to compare travelling reference instruments, that is instruments that are used to transfer calibrations. Brewers #017, managed by International Ozone Services (IOS) and directly calibrated to the Environment and Climate Change Canada, Toronto Triad (Fioletov et al., 2005; Fioletov and Netcheva, 2014; Netcheva, 2014), and #158, managed by Kipp & Zonen, manufacturer of the Brewer spectrophotometer, took part in the X RBCC-E campaign. Since 2007 the Brewer #158 is calibrated annually during the RBCC-E intercomparisons so it is already referenced to the RBCC-E triad. The calibration of IOS and RBCC-E Triads primary reference for the travelling instruments are discussed by León-Luis et al. (2018).

The agreement between the travelling reference instrument during this campaign was found to be quite good, with differences lower than 0.5% for OSC lower than 900 DU (see Table 2). Note that Brewer #017 is a single-monochromator instrument and is affected by stray light, thus underestimating the ozone at high OSC values above 600 DU.

Table 2 shows the comparison of the reference instruments during the RBCC-E campaigns and the corresponding calibration report. The agreement is generally around +/- 0.5% but with exceptions. In contrast to the RBCC-E travelling instrument which is transported by boat/car to Huelva and as hand luggage using two extra-sites of the plane, to Arosa campaigns. The travelling instrument are usually transported by cargo and can have issues during transportation that are reflected in table 2 or require instrumental changes for example #158 had a new PMT and new electronics during Arosa 2014 and the SL tests do not reflect this change.

### 3.3 Blind Days

A blind comparison with the reference Brewer instrument is performed at the beginning of the campaign, thus providing information on the initial status of the instrument, i.e. how well the instrument performs using the original calibration constants (those operational at the instrument's station). Possible changes of the instrument response due to the travel can be detected through the analysis of internal tests performed before and after the travel.

The analysis of the SL historical record is one of the principal tools to establish the stability of the instrument calibration. Moreover the comparison with a reference during calibration campaigns is the most suitable tool to determine if the observed R6 changes are related or not with changes in the ETC constant. During the El Arenosillo 2015 intercomparison campaign, most instruments agreed on average with the corresponding R6 reference value within $\pm 10$ units, which is about 1% in ozone. The stray light record tracks well small and slow changes on instrument responsivity but has issues when these changes are abrupt or huge. Generally, from our experience, the SL correction tracks the changes in the relative spectral sensitivity of the Brewers well unless the instruments characteristics change (whether intentionally or not) in ways that affect the ozone observations differently from the SL measurements, e.g. changes in the iris diaphragm, neutral density filters or zenith prism pointing towards the lamp or the sun. During the analysis we focus on the instruments that showed deviations of R6 values to the reference larger than 20 units (Fig. 11). Even in these cases for some instruments, for example Brewer#228, the SL correction improves the comparison, whereas for others like #163, where the change was produced by the modification of the

fore optics the opposite happens. The comparison with a standard reference instrument is the only way to assess whether the SL correction properly tracks changes on the calibration or the changes observed are just due to changes of the lamp's spectral emission (Fig. 10). This analysis will determine if a re-evaluation of the ozone observations between calibrations are required after an analysis of the history of the instrument.

On the following analysis we do not consider non-operative instruments, an operative instrument is one which is capable of providing reliable data during the campaign suitable for submission to databases like EUBREWNET or WOUDC (Word Ozone and UV Data Center). This is not the case of #151 because the instrument shows a huge change on the SL record from the last calibration (300 R6 units while the instrument range is +/- 60 UD), indicating serious instrumental issues.

Table 3 shows the mean relative difference for the simultaneous direct sun measurements with the reference for all the
participating instruments, with and without the standard lamp correction, in the stray light-free OSC region. With the exception of Brewer #151, that can not be considered an operational instrument, the maximum difference found is 1.5%. This is a really good result considering that most of the instruments were calibrated two years previously. The third column of the table shows the average of the best result for all the observation OSC range. This result is an estimation of the calibration agreement of the EUBREWNET network, with half of the instruments showing a perfect agreement within ±0.5%, and 75% within the ±1%
level.

## 3.4   Final comparison

We define the final days as those available after the maintenance work has been finished for each participating instrument. These days are used to calculate the final calibration constants, so we endeavour not to manipulate the instruments during this period. Furthermore, the SL R6 value recorded during the final days is normally adopted as the new reference value. It is also
expected that this parameter will not vary more than 5 units during this period. We show in Fig. 12 the differences between the daily standard lamp R6 ratio and the proposed R6 reference value during the final days. As expected, the recorded SL values did not vary more than 5 units during this period.

Deviations of ozone values for all the participating instruments with respect to the RBCC-E travelling standard Brewer #185 are shown in Fig. 13 and summarized in Table 3. We have recalculated the ozone measurements using the final calibration
constants and, in the case of single Brewer instruments, with and without the stray light correction as described in Sect. 2.

The effect of the stray light correction is not large on the statistics, only 30% (sza> 60º) of the observations are affected on the single Brewer, and only for %15 (sza>70º) is the effect bigger than 1% but taking into account the stray light allows for the instruments to be calibrated using the one parameter method. Not taking into account this correction using the two parameters method as in Bojkov et al. (2008) can cause a misleading calibration constant. For all the instruments, both the one parameter
and the two parameters ETC transfer methods agreed to each other within the limit of ±5 units for ETC constants and ±0.3% for ozone absorption coefficients, which is an indication of the quality of the calibration provided.

We achieved a good agreement with the reference instrument Brewer #185 using the final calibration constants, see Fig. 13 and Table 3. With the application of the stray light correction to the single Brewer spectrophotometers, all instruments are within the ±0.5% agreement range.

## 4 Conclusions

To summarise the calibration results of the 10th RBCC-E campaign, we found that during the blind days, using the two-year-old calibration issued in the previous campaign,

- 16 Brewer spectrophotometers (∼75% of the participating instruments) were within the $1\%$ agreement range.

- 10 Brewer spectrophotometers (∼50%) were within the $\pm 0.5\%$ range, i.e., show a perfect agreement.

- The max average error was 1.5% for operational Brewer instruments within stray-light-free conditions (OSC < 700 DU).

This results are in agreement with the RBCC-E campaigns celebrated in Huelva and Arosa from 2009 to 2015 (Figure 14), in this period 85 spectrometers have been calibrated: 59 (69%) show an agreement better than 1%, 32 (38%) within $0.5\%$ and 7 (8%) show a discrepancy greater than 2%.

10     For all participating instruments are calibrated with one parameter calibration. One parameter and the two parameters ETC transfer methods agreed to each other within the limit of $\pm 5$ units for ETC constants and $\pm 0.3\%$ for ozone absorption coefficients, indicating a high quality calibration.

After the new calibration was issued at the end of the X RBCC-E campaign,

- All participating Brewer spectrophotometers were within the $\pm 0.5\%$ agreement range.

- Without the stray light correction implemented large errors of up to 4% can be expected for single-monochromator Brewer instruments operating at OSC larger than 1000 DU.

- The implementation of the stray light correction in the calibration of single Brewer instruments improved their performance. Therefore, this correction has been introduced in Eubrewnet for the automatic processing of data sent by single monochromators Brewers.

20     *Acknowledgements.* All this work would have not been possible without the participation, work and dedication of all the Brewer operators in the RBCCE intercomparison campaigns.

This article is based upon work from COST Action 1207 "EUBREWNET", supported by COST (European Cooperation in Science and Technology). This work has been supported by the European Metrology Research Programme (EMRP) within the joint research project ENV59 "Traceability for atmospheric total column ozone" (ATMOZ). The EMRP is jointly funded by the EMRP participating countries
25     within EURAMET and the European Union. We also gratefully acknowledge further support by the Fundación General de la Universidad de La Laguna. This study and the campaigns were supported at large part by ESA project CEOS Intercalibration of ground-based spectrometers and lidars (ESRIN contract 22202/09/I-EC).

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

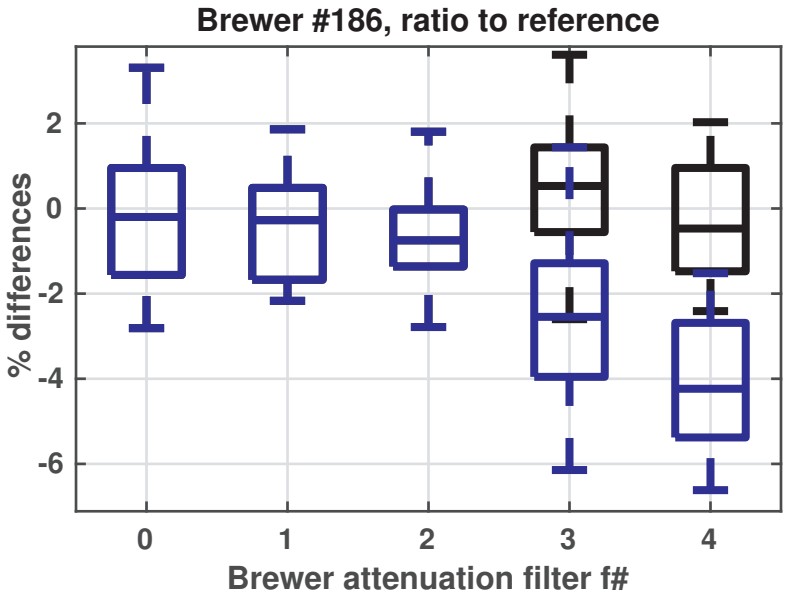

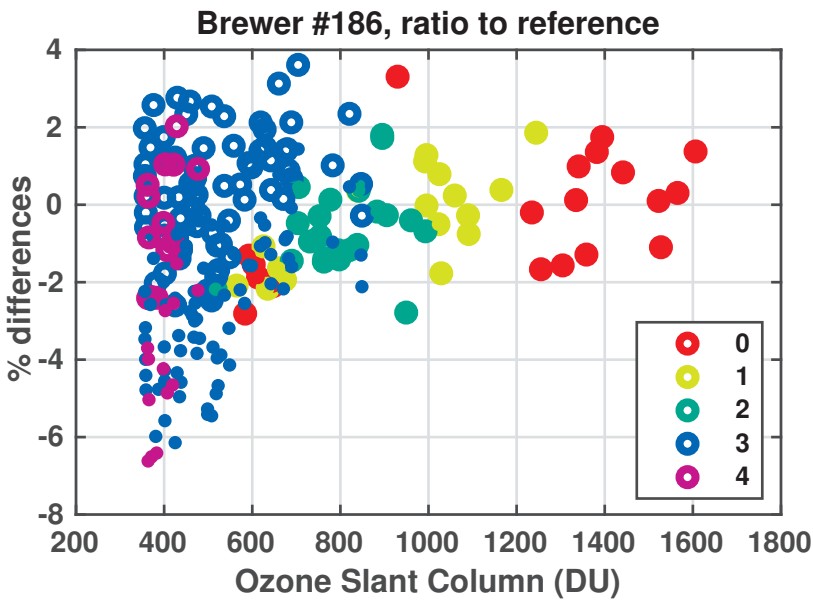

**Figure 6.** BoxPlot of the percentual differences with respect to the reference grouped by filter for Brewer #186, in blue without correction, and in black after applying the correction to filters 3 and 4 (upper panel). On the lower panel percentage differences with respect to the reference grouped by filter, without correction (solid dots), and after the application of the correction to filters 3 and 4 (open circles). Colors indicate the number of the filter; see the legend.

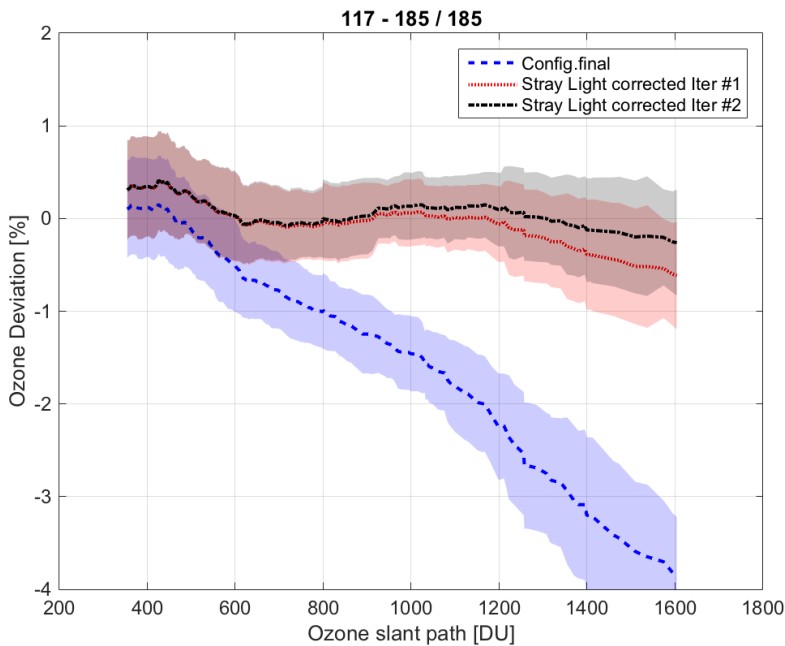

**Figure 7.** Percentage ozone differences with respect to the reference *vs.* Ozone Slant Path. In blue, using the final configuration constants, and in black and red, after the stray light correction has been applied, with one and two iterations, respectively. Data are averaged in ±50DU intervals, the shadow area represents one standard deviation.

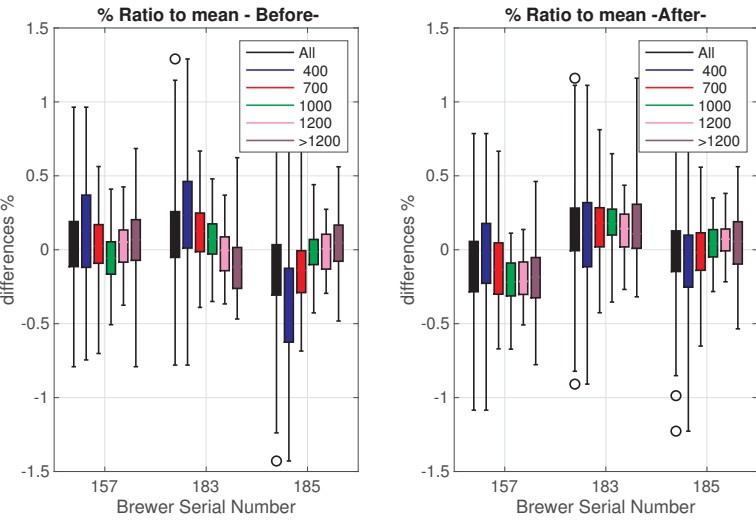

**Figure 8.** Box plot of the ozone percentage deviation from the mean of the RBCC-E triad reference Brewer#157, Brewer#183 and Brewer#185 before (left panel) and after (right panel) the X RBCC-E campaign at El Arenosillo in 2015, grouped by ozone slant columns ranges. The color indicates the intervals used for the averaging of the observations- blue, lower than 400 DU; red, between 400 and 700DU; green, between 700 and 1000DU; pink, between 1000 and 1200DU; and purple for OSC >1200 DU. In black the average of all observations.

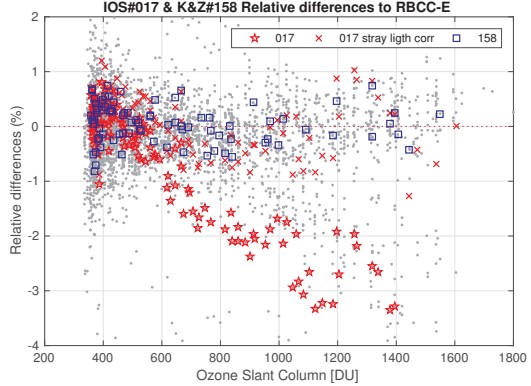

**Figure 9.** Comparison of reference instruments during the X RBCC-E campaign: relative differences with respect to the IZO reference using the initial configuration during the campaign, in red for the IOS Brewer #017 (stars are used for the original observations and crosses for the stray light corrected ones), and in blue for the K&Z Brewer #158. The gray points are the relative differences to the IZO reference for all participating instruments

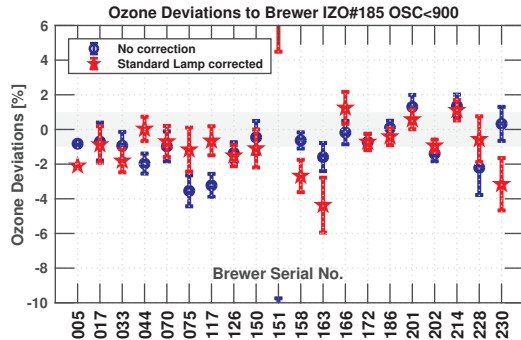

**Figure 10.** Percentage mean difference for the simultaneous direct sun measurements with the reference for all the participating instruments, with and without the standard lamp correction, in the stray light-free OSC region (OSC<900). (The Brewer #151 is an not operative brewer , is not providing reliable data , and is outside the limits.)

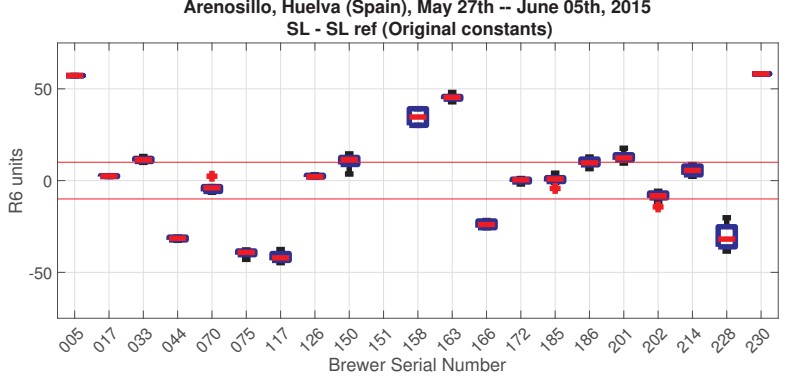

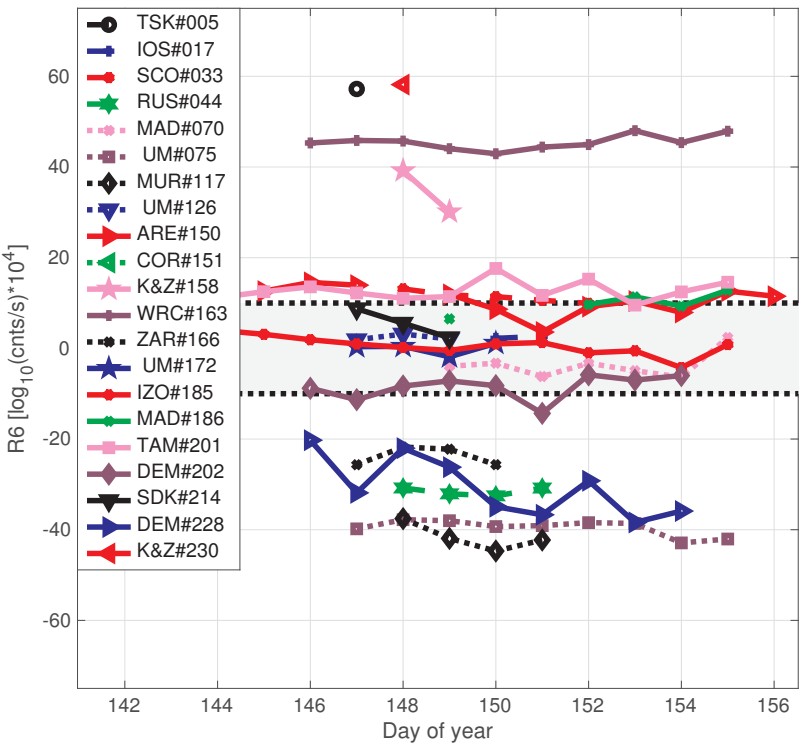

**Figure 11.** Standard lamp R6 difference with respect to the R6 reference value from the last calibration during the blind days, before the maintenance, the upper panel shows the mean value for each instrument and the lower panel the daily mean during the campaing. Variations within the $\pm 10$ units range ($\sim$1% in ozone) from the reference value are considered normal, whereas larger changes would require further analysis of the instrument performance.

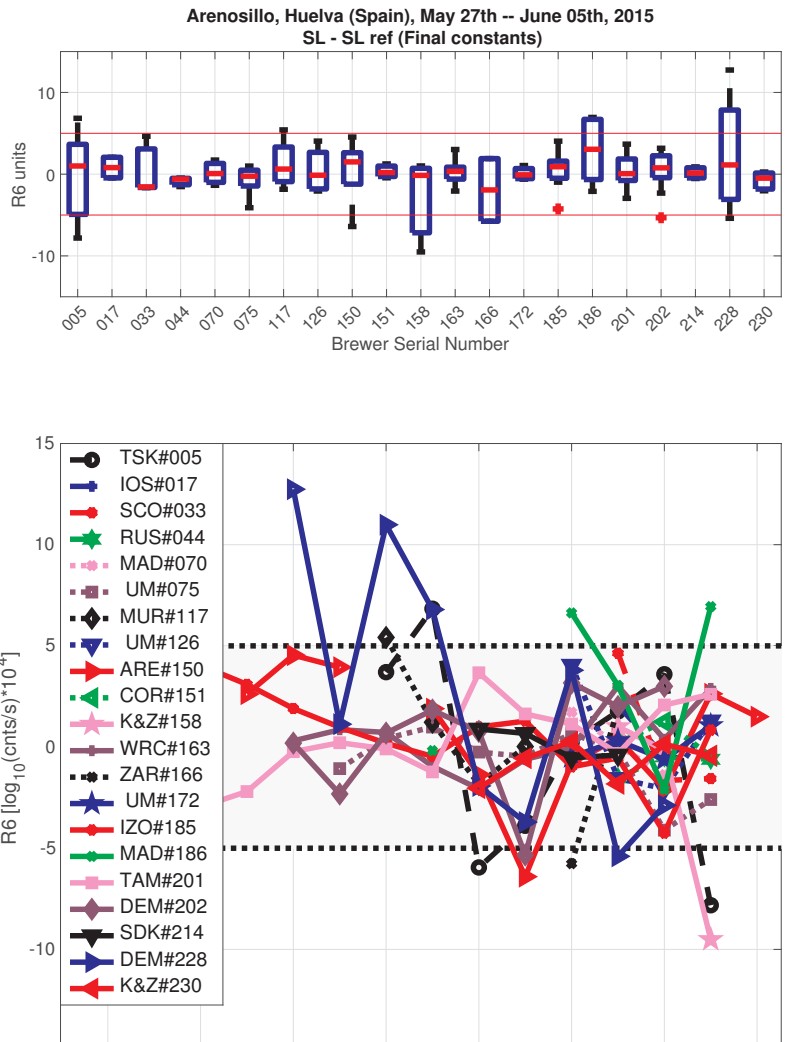

**Figure 12.** Differences between the daily standard lamp R6 ratio and the proposed R6 reference value during the final days, the upper panel shows the mean value for each instrument, and the lower panel the daily mean during the campaign, is important to get an stable R6 value during the final days , within the $\pm 5$ units range ($\sim$.5% in ozone), to stabilise the reference value of the calibration.

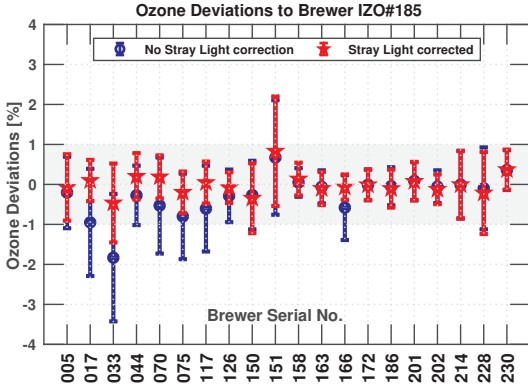

**Figure 13.** Final days mean percentage difference with respect to the reference Brewer for the simultaneous direct sun measurements for all the participating instruments, blue circles shows results without the stray light correction and red starts show results with the correction applied to single Brewer spectrophotometers.

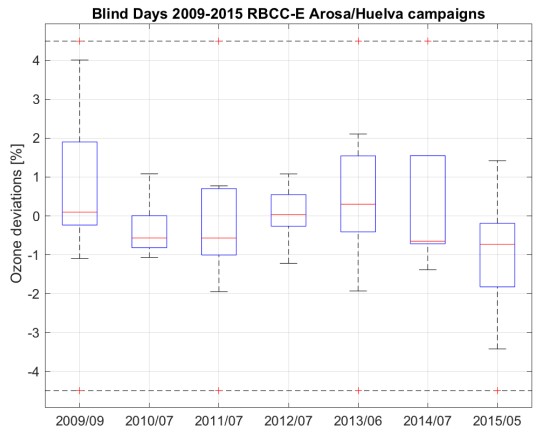

**Figure 14.** Ozone deviations for the Blind Days with respect to the reference Brewer for the simultaneous direct sun measurements for all the participating instruments during the RBCC-E regular campaigns 2009-2015, the campaigns performed in odd years correspond to Arosa (Switzerland) and in even years in Huelva (Spain). This results correspond to the stray-light-free region OSC< 700 DU, the outliers (red cross at +/- 4.5% levels) generally correspond to non-operating instruments.

**Table 1.** Principal Investigators and Instruments participating on the X RBCC-E campaign

| Nr. | Country | Brewer | Participants | |
|---|---|---|---|---|
| 1 | Greece | 005 | Alkis Bais | Thessaloniki University |
| 2 | Canada | 017 | Volodia Savastiouk | International Ozone Services |
| 3 | Spain | 033 | Juan R. Moreta | AEMET,State Meteorological Agency from Spain |
| 4 | Russia Federation | 044 | Vadim Shirotov | Scientific and Production Association "Typhoon" |
| 5 | Spain | 070 | Juan R. Moreta | AEMET,State Meteorological Agency from Spain |
| 6 | United Kingdom | 075 | John Rimmer | Manchester University |
| 7 | Spain | 117 | Juan R. Moreta | State Meteorological Agency from Spain |
| 8 | United Kingdom | 126 | John Rimmer | Manchester University |
| 9 | Spain | 150 | J. M. Vilaplana | National Institute for Aerospace Technolog |
| 10 | Spain | 151 | Juan R. Moreta | State Meteorological Agency from Spain |
| 11 | Netherlands | 158 | Oleksii Marianenko | Kipp & Zonen |
| 12 | Switzerland | 163 | Julian Gröebner | Physikalisch-Meteorologisches Observatorium Davos |
| 13 | Spain | 166 | Juan R. Moreta | AEMET, State Meteorological Agency from Spain |
| 14 | United Kingdom | 172 | John Rimmer | Manchester University |
| 15 | Spain | 185 | Alberto Redondas | Izaña Atmospheric Research Center,AEMET |
| 16 | Spain | 186 | Juan R. Moreta | AEMET,State Meteorological Agency from Spain |
| 17 | Algeria | 201 | Bukelia Lamine | National Meteorological Office |
| 18 | Denmark | 202 | Paul Eriksen | Danish Meteorological Institute, |
| 19 | Finland | 214 | Tomi Karpprinen | Finnish Meteorological Institute |
| 20 | Denmark | 228 | Niss Jepsen | Danish Meteorological Institute, |
| 21 | Netherlands | 230 | Keith M. Wilson | Kipp & Zonen |

**Table 2.** Reference Comparison during RBCC-E campaigns, the Brewer #017 is the travelling reference from International Ozone Service (IOS), the Brewer #158 is the travelling reference from Kipp & Zonen and finally Brewer #145 from Environmental Canada is a double Brewer and direct calibrated to the World Reference Triad who participates on the previous RBCC-E campaigns

| Location | year | #017 | #158 | #145 | Report |
|----------|------|------|------|------|--------|
| Arosa  | 2008 | -0.6 |      |      | (Redondas and Rodriguez-Franco, 2008) |
| Huelva | 2009 | -0.6 | 0.8  | -0.1 | (Roozendael et al., 2012) |
| Arosa  | 2010 | -0.6 |      |      | (Roozendael et al., 2013b) |
| Huelva | 2011 | -0.1 | -0.2 | -0.6 | (Roozendael et al., 2013a) |
| Arosa  | 2012 |      | -0.1 |      | (Redondas et al., 2015) |
| Huelva | 2013 | -1.0 | 0.7  |      | (Redondas and Rodriguez-Franco, 2015a) |
| Izaña  | 2014 |      |      | -2.2 | (Redondas et al., 2014b) |
| Arosa  | 2014 | -1.2 | 1.5  |      | (Redondas and Rodriguez-Franco, 2015b) |
| Huelva | 2015 | -0.5 | -0.5 |      | this work |

**Table 3.** Summary of mean percentage difference before calibration, without and with Standard Lamp Correction, and after the calibration, on the last column with the stray light correction applied.

| Brewer ID | No corr. | SL corr. | Blind | Final | Stray |
|:---:|:---:|:---:|:---:|:---:|:---:|
| **005** | - | - | -1.93 | -0.2 | -0.08 |
| **017** | -0.31 | -0.49 | -0.98 | -0.95 | 0.11 |
| **033** | -0.8 | -1.77 | -1.09 | -1.83 | -0.48 |
| **044** | -2.04 | 0.13 | -0.21 | -0.27 | 0.2 |
| **070** | -0.73 | -0.42 | -0.71 | -0.53 | 0.18 |
| **075** | -3.42 | -0.71 | -1.2 | -0.8 | -0.2 |
| **117** | -3.38 | -0.45 | -0.68 | -0.6 | 0.04 |
| **126** | -1.25 | -1.41 | -1.36 | -0.29 | -0.08 |
| **150** | -0.45 | -1.07 | -0.45 | -0.27 | - |
| **151** | -17.36 | 9.94 | 7.95 | 0.67 | 0.83 |
| **158** | -0.54 | -2.45 | -0.54 | 0.05 | - |
| **163** | -1.5 | -4.16 | -1.5 | -0.06 | - |
| **166** | -0.15 | 1.45 | -0.24 | -0.58 | - |
| **172** | -0.67 | -0.67 | -0.67 | -0.01 | - |
| **186** | 0.13 | -0.34 | 0.13 | -0.05 | - |
| **201** | 1.21 | 0.52 | 0.52 | 0.09 | - |
| **202** | -1.39 | -0.95 | -0.95 | -0.06 | - |
| **214** | 1.42 | 1.19 | 1.19 | -0.01 | - |
| **228** | -1.93 | -0.4 | -0.4 | -0.1 | - |
| **230** | -0.15 | -3.48 | -0.15 | 0.36 | - |