# Peer review of "EUBREWNET RBCC-E Huelva 2015 Ozone Brewer"

_Atmospheric Chemistry and Physics, 2017_

## Referee Comment (RC1) · Anonymous Referee #1 · 29 Jan 2018

General comments:

The content of the paper is very interesting and contributes to the understanding of shortcomings in the total ozone observations, especially of single Brewers, and presents possible corrections to improve data measured at low sun. In principle the publication is recommended after some revisions.

- One major deficit is that SO2-calibration is not discussed. The effect of SO2-values on the comparison with Dobsons and also between the different Brewers is important for the interpretation of the results. As far as I know the SO2-values of BR017 in clear air is around Zero, whereas Brewers calibrated against the Tenerife triad show up to -5 DU equivalent SO2 and subsequently higher TOC values. This leads to misinterpretation when only TOC is looked at.

[Figure]

- Only the influence of internal stray light (unwanted radiation of "wrong" wavelengths inside the instrument) is discussed, but the external one (scattered sun light from the sky around the sun disc) is not mentioned and discussed as a source of similar effects.

Specific comments:

- p1, line 6 and p3, line 7/8: Davos as location for the WRC-UV should be mentioned; the exact name is World Calibration Center – Ultraviolet Section (WCC-UV) at the Physikalisch-Meteorologisches Observatorium Davos / World Radiation Center (PMOD/WRC).

- p2, line 8: it is not easy to find the correct name of the mentioned SAG, but in any case ozone should be added: "WMO/GAW Scientific Advisory Groups (SAG) on Ozone" as proposal.

- p5, section 2.1.: Stray light effect should be distinguished between internal stray light, which means unwanted measured radiation in other wavelengths inside the instrument (double Brewers show a better stray light suppression than single Brewers) and external stray light: scattered sky light around the sun disc with a different spectral composition than direct sun light. This external stray light also leads to a drop in ozone values at lower sun depending on the turbidity (aerosol amount and or haze) and the instrument's field of view (similar effect with single and double Brewers). This is one reason why the TOC measured with Dobsons with their wider FOV drops earlier than TOC even from single Brewer, although this old sepectrometer is a double monochromator with relatively small amount of internal stray light.

- p5, line 26: OSC is the product of TOC and relative slant path through the ozone layer mue and not the airmass, which are significantly different at low sun.

- p5, line 28: in this context the statement "For this type of Brewer, only the stray-light-free region is used to determine the ETC, which generally ranges from 300 to 900 DU in the OSC, depending on the instrument." is a little bit misleading. The given maximum

of 900 DU of a stray light free region of single Brewer seems to be very low. It means, that the single Brewer TOC of 300 DU already drops when a mue-value of 3 is reached, which should not be the case under normally clear sky condition for normal Brewers. In this special case an OSC of 600 for BR 117 shows a very bad instrument with strong internal stray light effect. This should be mentioned explicitely.

- p6, line 5, an amendment with the word "empirically" before the word "corrected" would make it clearer, that it is not a physically based correction.

- p8, line 5-6: the statement that BR017 is underestimating ozone at high OSC above 600 DU seems to be too strong, although it is shown in Fig. 8, which is, however, not mentioned in the text. In my opinion (see also second last comment) 600 DU represents very small mue-values of around 2 at normal TOC of 300 DU. These mue-value is not common for TOC-drops of single Brewer observations after my experience. Perhaps it would be helpful to present a graph for different Brewers (reference, single and double) showing the daily course of TOC with mue as x-axis.

- p8, line 13: the SL test is not an ozone measurement, as there is definitely no amount of ozone between the lamp and the PMT. It is a check of the spectral response as mentioned in line 14.

- p9, Figure 6 and p10, Figure 7: no explanation is given in the text of the caption for the numbers of the boxes. For an insider it is clear that the rel. deviation in the different OSC bins is mentioned. In Figure 7 some blue and red circles are drawn. What do they mean? - p11, Figure 9: BR151 is outside the range.

- p12, Figure 10: is not very clear, the difference between the two panels is not explained; are there differences between the captions for the y-axis?

- p15, Figure 13 caption: in even years correspond to Arosa and in odd years in Huelva.

---

## Referee Comment (RC2) · Anonymous Referee #2 · 5 Feb 2018

General comments

The paper "EUBREWNET RBCC-E Huelva 2015 Ozone Brewer Intercomparison" by Redondas et al. describes some of the main findings from an international comparison campaign of Brewer spectrophotometers. After an introduction about the Brewer ozone retrieval algorithm and the calibration transfer techniques, particular attention is given to an empirical parametrisation/correction of stray light applied to the single-monochromator instruments. A short discussion about the "standard lamp correction" to track the radiometric stability of the spectrophotometers is also provided.

In my opinion, the paper potentially raises the following important questions:

1. what is the maximum attainable reproducibility by well-calibrated Brewer spectropho-

tometers?

2. what are the most common sources of error/instability in the Brewer measurements? How can they be identified and solved during an intercomparison?

3. how important is the stray light effect on ozone estimates and what techniques can be used to overcome this issue?

4. how good is the agreement among reference instruments used to calibrate the Brewer network?

Therefore, the manuscript, in principle, addresses relevant scientific questions within the scope of ACP. However, I have two main concerns related to the paper:

1. the stray light and the standard lamp corrections should be discussed more properly (cf. Specific comments);

2. the manuscript resembles more to a technical report than a scientific article (especially considering that the manuscript has been submitted to ACP). The previously listed scientific questions (1-4) deserve a deeper discussion, and should be better enhanced (e.g., they should be presented in the introduction, together with a set of bibliographic references, and answered in the main text through quantitative results). Technical details (e.g., determination of the dead time, dispersion function, etc.) that would be suitable for a report should be omitted in the present paper if not relevant to the scientific discussion. A reorganisation of the paper, keeping theory and results better apart, would improve readability (cf. Technical corrections).

Once these remarks are properly addressed, the paper can be published in ACP.

Specific comments

1. It is rather trivial that the comparison between single- and double-monochromator instruments improves when a stray light empirical correction, obtained from the comparison itself, is applied back to the same set of data. What is not obvious, in my

opinion, is that the correction obtained during the intercomparison can be also used to improve accuracy when the Brewer is moved back to the home institution after the campaign. This would be an important conclusion, but some points should be addressed:

A. it should be proved (or discussed) that the correction only depends on the instrumental characteristics and not on the measurement site;

B. stray light should be characterised, during the intercomparison, for the full range of slant ozone values reached during normal operation. This is particularly important for single-monochromator instruments located at high-latitude stations. Can the authors state that the OSC range during the intercomparison is wide enough?

C. the authors affirm that "These parameters, determined in several campaigns, have been found to be stable" (page 6 l. 8). This is a key point: can they show some quantitative data demonstrating that the correction is stable over time?

2. The section about the SL correction (page 9) is quite inconclusive. It is demonstrated that the SL correction does not improve the accuracy for some Brewers (while it does for others) and that the only way to verify it is an intercomparison against a reference instrument. In that case, how should the Brewer data be reprocessed from one intercomparison to the next? My concern is not to spread the idea that the Brewer data quality is aleatory and that the community does not know how to reprocess the data for improving their quality.

Technical corrections

page 1 l. 4-6: omit the reference to UV and QASUME in the abstract if the UV results are not discussed in the text;

page 1 l. 4: "Twenty-one". It could be useful to mention already in the abstract how many single- and double-monochromator instruments have been studied;

page 1 l. 7: at the first occurrence, use "spectral stray light" instead of only "stray light", to distinguish it from other sources of stray light (e.g., multiple scatter stray light n the

field of view);

page 1 l. 9-10: omit 76% and 50% percentages (16/21 is easy to calculate, and 10/21 is 47%, not 50%);

page 1 l. 10: state the air mass range relative to the 1% and 0.5% thresholds;

page 1 l. 11: enhance the outcomes of the paper, e.g. why / to whom those findings are important?

page 2, Sect. 1: I would expect a more sound introduction, focussing on scientific issues (and related literature, e.g. on the stray light effect, etc.), rather than a summary of the previous campaigns. Also, since the previous technical reports are mentioned, it should be specified what the novelty of the present study is;

page 2 l. 17: "Figure" instead of "Fig." at the beginning of a sentence (https://www.atmospheric-chemistry-and-physics.net/for_authors/manuscript_preparation.html);

page 3 l. 2: table 2 is cited before table 1, please reverse the order of tables;

page 3 l. 3: the measurement site should be better described, since the local characteristics of the measurement site impact of the results of the campaign;

page 3 l. 4-5: since UV radiation is out of the scope of this paper, omit the references to QASUME and WRC-UV;

page 3 l. 8-21: those paragraphs have nothing to do with Sect. 1.1, entitled "The X RBCC-E campaign". Omit them and reference the EUBREWNET paper (https://www.atmos-chem-phys-discuss.net/acp-2017-1207/) in your Introduction;

page 4 Eq. 1: if alpha is defined as in Eq. 3, Eq. 1 should read (ETC-F)/(alpha*mu) instead of (F-ETC)/(alpha*mu). Otherwise, if alpha is defined negative (as in the Brewer literature), Eq. 3 should have a minus sign;

page 4 l. 8: define what "double ratios" are, for the unexperienced reader. "corrected for the Rayleigh effects" -> "to which the effect of Rayleigh scattering has been subtracted (Eq. 2)";

page 4 l. 16: "verify" -> "satisfy";

page 4 Eq. 5: Eq. 6 is explained in the following text, please explain Eq. 5 as well;

page 5 l. 1: "calibration" -> do you rather mean "characterization"?

page 5 l. 2: "filter attenuation" -> has any filter already been mentioned in the text?

page 5 l. 3: "The wavelength calibration allows to..." -> "An accurate determination of the operational wavelengths is needed to". "are" -> "is";

page 5 l. 6: "Finally, the ETC transfer is performed by comparison..." -> "Finally, the ETC must be determined by transfer from a reference Brewer (Sect. 2.1) or, in the case of the reference instruments, by the Langley method" (include bibliographic references about the Langley method);

page 5 l. 8-9: "changes ... wavelength calibration will affect the final ETC and changes in the wavelength calibration will affect also the final ETC" -> is there any difference between the two sentences?

page 5 l. 18: "calibration" -> "radiometric";

page 5 l. 20: "for simultaneous measurements BY BOTH INSTRUMENTS";

page 5 Eq. 7: explain that "i" does not refer to wavelength, as in Eqs. 2-6, but to the sample (better: use a different subscript). Correct the sign "-" to "+" according to Eq. 2 (cf. my previous comment);

page 5 l. 27: define the "stray-light free region" from a quantitative point of view;

page 6 Fig. 3: Figs. 3 and 4 are very similar. Consider keeping only one of them. Add units of ozone slant path (cm STP);

page 6 Eq. 8: the "F0" notation of Eq. 4 may be confused with "Fo" of Eq. 8. Consider using a different notation;

page 6 Eq. 8 and 9: as far as I understand, the k(X*mu)ˆ2 term can be attributed to either the F's or to the ETC's, but not to both of them at the same time. In this case, the text should be clearer;

page 6 line 7: "stray-light free OSC region" -> this expression is misleading, since the k(X*mu)ˆs cannot be zero for mu>=1 (or reformulate the correction term to be zero for mu=1). Also, could the ETC be directly retrieved from the fit, together with k and s?

page 7 Eq. 10: since k is negative, if we want the ozone to increase at the next iteration, there must be a "-" sign, unless alpha is defined negative (Eq. 3). Please, clarify;

page 7 line 5: "2000 DU" -> to what air mass does it correspond, in El Arenosillo? Is the calculation of the air mass reliable at this SZA? Notice that the stray light correction presented in the paper depends on the product X*mu: if X is low during the determination of the correction, mu must be large (but still accurate!) to cover a sufficiently wide range of OSC, since, at high latitudes, the same OSC can be reached at lower mu's;

page 7 "3.1 Reference Calibration" -> or "Agreement between reference instruments"?

page 8 l. 1-6: the comparison of the reference instruments is an interesting topic, and the RBCC campaigns a very unique chance to investigate it. However, the authors should discuss it with more details. E.g., how are the ECCC Toronto Triad and #158 calibrated? The procedure is explained for the IZO triad, but not for other reference instruments;

page 8 l. 5-6: why two different thresholds (900 DU and 600 DU)?

page 8 l.8 to page 9 l. 7: this basically theoretical part should be described in the "Methods" section, not in the "Results";

[Figure]

page 9 l. 13: Brewer #165 does not exist in the list of the instruments;

page 10 l. 3: "Brewer #151 can not be considered an operational instrument" -> what does this sentence mean?

page 10 l. 4: "ago" -> "before";

page 10 l. 12 and 14: "more than 5 units during this period" repeated twice;

page 11 Fig. 9: ordering the x-axis by serial number would improve readability of the chart;

page 12 Fig. 10: both the bias and the spread for Brewers 158 and 228 is large (in the boxplot). Is there any connection between bias and spread (e.g., instability of the lamps...)?

page 12 l. 1-5: move these lines to a "Methods" section;

page 13 l. 4: if the discussed results were representative of the overall Brewer network, this would mean that about 25% of the instruments are "out of spec" after two years. This is a strong statement: could the author elaborate on this?

page 13 l. 7: "celebrated" -> "taking place"?

page 14: can the authors draw more definitive and general conclusions from the study?

page 15 Fig. 13: could the variations over time be explained by the variability of the participating instruments? This should be mentioned in the text. Caption: what does "no operating" mean?

---

## Author Comment (AC1) · 9 May 2018

**EUBREWNET RBCC-E Huelva 2015 Ozone Brewer Intercomparison**

Alberto Redondas[1,2], Virgilio Carreño[1,2], Sergio F. León-Luis[1,3], Bentorey Hernández-Cruz[2,3], Javier López-Solano[1,2,3], Juan J. Rodriguez-Franco[1,3], José M. Vilaplana[4], Julian Gröbner[5], John Rimmer[6], Alkiviadis F. Bais[7], Volodya Savastiouk[8], Juan R. Moreta[9], Lamine Boulkelia[10], Nis Jepsen[11], Keith M. Wilson[12], Vadim Shirotov[13], and Tomi Karppinen[14]

[1]Izaña Atmospheric Research Center, Agencia Estatal de Meteorología, Tenerife, Spain
[2]Departamento de Ingeniería Industrial, Universidad de La Laguna, Tenerife, Spain
[3]Regional Brewer Calibration Center for Europe, Izaña Atmospheric Research Center, Tenerife, Spain
[4]National Institute for Aerospace Technology - INTA, Atmospheric Observatory "El Arenosillo", Huelva, Spain
[5]Physikalisch-Meteorologisches Observatorium Davos/World Radiation Center, Davos, Switzerland
[6]Manchester University, Manchester, United Kingdom
[7]Laboratory of Atmospheric Physics, Aristotle University of Thessaloniki, Thessaloniki, Greece
[8] International Ozone Services, Toronto, Canada
[9]Agencia Estatal de Meteorología, Madrid, Spain
[10]National Meteorological Office, Algeria
[11]Danish Meteorological Institute, Copenhagen, Denmark
[12]Kipp & Zonen, Delft, The Netherlands
[13]Scientific and Production Association "Typhoon", Obninsk, Russia
[14]Finnish Meteorological Institute, Sodankyla, Finland

*Correspondence to:* Alberto Redondas (aredondasm@aemet.es)

**Response to Reviewer #1**

**Comment:** One major deficit is that SO2-calibration is not discussed. The effect of SO2-values on the comparison with Dobsons and also between the different Brewers is important for the interpretation of the results. As far as I know the SO2-values of BR017 in clear air is around Zero, whereas Brewers calibrated against the Tenerife triad show up to -5 DU equivalent SO2 and subsequently higher TOC values. This leads to misinterpretation when only TOC is looked at.

**Answer:** The work is focused on the ozone calibration and our work during these years, unfortunately we don't perform SO2 transfer and use the Software from IOS (International Ozone Services) to derive it (**?**). However in some cases we perform the SO2 calibrations performed by the RBCC-E as is discussed in the supplement. In this work we don't find significant differences between the Langley Calibration and the "zero calibration" usually performed by the International Ozone Service traveling #017, but we cannot find the 5 DU differences **?**.

We agree with the referee that is an important issue to address in future intercomparsions and also need some review concerning the SO2 cross sections and the validity of the assumptions of the ozone/so2 coefficient used on the brewer.

**Comment:** Only the influence of internal stray light (unwanted radiation of "wrong" wavelengths inside the instrument) is discussed, but the external one (scattered sun light from the sky around the sun disc) is not mentioned and discussed as a source of similar effects.

**Answer:** We have introduced this source of stray-light in the paper and a more detailed description of it

5 **Comment:**

- p1, line 6 and p3, line 7/8:li Davos as location for the WRC-UV should be mentioned; the exact name is World Calibration Center – Ultraviolet Section (WCC-UV) at the Physikalisch-Meteorologisches Observatorium Davos / World Radiation Center (PMOD/WRC).

**Answer:** Done

10 **Comment:** - p2, line 8: it is not easy to find the correct name of the mentioned SAG, but in any case ozone should be added: "WMO/GAW Scientific Advisory Groups (SAG) on Ozone" as proposal.

**Answer:** done

**Comment:** p5, section 2.1.: Stray light effect should be distinguished between internal stray light, which means unwanted measured radiation in other wavelengths inside the instrument (double Brewers show a better stray light suppression than

15 single Brewers) and external stray light: scattered sky light around the sun disc with a different spectral composition than direct sun light. This external stray light also leads to a drop in ozone values at lower sun depending on the turbidity (aerosol amount and or haze) and the instrument's field of view (similar effect with single and double Brewers). This is one reason why the TOC measured with Dobsons with their wider FOV drops earlier than TOC even from single Brewer, although this old sepectrometer is a double monochromator with relatively small amount of internal stray light.

20 **Answer:** We add this distinction in the revised version of the paper.

**Comment:** - p5, line 26: OSC is the product of TOC and relative slant path through the ozone layer mue and not the airmass, which are significantly different at low sun.

**Answer:** the correction is introduced.

**Comment:**

25 p5, line 28: in this context the statement "For this type of Brewer, only the stray-light- free region is used to determine the ETC, which generally ranges from 300 to 900 DU in the OSC, depending on the instrument." is a little bit misleading. The given maximum of 900 DU of a stray light free region of single Brewer seems to be very low. It means, that the single Brewer TOC of 300 DU already drops when a mue-value of 3 is reached, which should not be the case under normally clear sky condition for normal Brewers. In this special case an OSC of 600 for BR 117 shows a very bad instrument with

30 strong internal stray light effect. This should be mentioned explicitly.

**Answer:** , we agree that this fact has to be mentioned, we have choose the #117 because with also the #017 and the

[Figure]

**Figure 1.** Ratio to the reference of the single brewers duringt the campaign the upper x-axis indicate the solar zenith angle with the assumption of a constant Ozone of 300 DU.

have the strongest stray light, we have added the corresponding information, but we cannot consider the 117 as a bad instrument and is not an outlier (See Figure below) .

**Comment:** - p6, line 5, an amendment with the word "empirically" before the word "corrected" would make it clearer, that it is not a physically based correction.

5    **Answer:** done

**Comment:** p8, line 5-6: the statement that BR017 is underestimating ozone at high OSC above 600 DU seems to be too strong, although it is shown in Fig. 8, which is, however, not mentioned in the text. In my opinion (see also second last comment) 600 DU represents very small mue-values of around 2 at normal TOC of 300 DU. These mue- value is not common for TOC-drops of single Brewer observations after my experience. Perhaps it would be helpful to present a graph for

10    different Brewers (reference, single and double) showing the daily course of TOC with mue as x-axis.

**Answer:** Agree that 600 DU is too low, but it is in agreement with what we found in previous campaigns. We will add a plot with the comparison in time and mu x axis for the single brewer of the campaign.

**Comment:** - p8, line 13: the SL test is not an ozone measurement, as there is definitely no amount of ozone between the lamp and the PMT. It is a check of the spectral response as mentioned in line 14.

15    **Answer:** The SL is a test but it use the same instrumental configuration as the ozone measurements and use the same ratio. The sentence is rephrased to avoid confusion.

**Comment:**

- p9, Figure 6 and p10, Figure 7: no explanation is given in the text of the caption for the numbers of the boxes. For an insider it is clear that the rel. deviation in the different OSC bins is mentioned. In Figure 7 some blue and red circles are drawn. What do they mean? - p11, Figure 9: BR151 is outside the range.

**Answer:** A description is included of the plots, the brewer 151 is outside the range but is not an operative instrument as is now described on the text.

**Comment:** - p12, Figure 10: is not very clear, the difference between the two panels is not ex- plained; are there differences between the captions for the y-axis?

**Answer:** A better description of the plot is included on the caption.

**Comment:** - p15, Figure 13 caption: in even years correspond to Arosa and in odd years in Huelva.

**Answer:** corrected

**References**

---

## Author Comment (AC2) · 9 May 2018

**EUBREWNET RBCC-E Huelva 2015 Ozone Brewer Intercomparison**

Alberto Redondas[1,2], Virgilio Carreño[1,2], Sergio F. León-Luis[1,3], Bentorey Hernández-Cruz[2,3], Javier López-Solano[1,2,3], Juan J. Rodriguez-Franco[1,3], José M. Vilaplana[4], Julian Gröbner[5], John Rimmer[6], Alkiviadis F. Bais[7], Volodya Savastiouk[8], Juan R. Moreta[9], Lamine Boulkelia[10], Nis Jepsen[11], Keith M. Wilson[12], Vadim Shirotov[13], and Tomi Karppinen[14]

[1]Izaña Atmospheric Research Center, Agencia Estatal de Meteorología, Tenerife, Spain
[2]Departamento de Ingeniería Industrial, Universidad de La Laguna, Tenerife, Spain
[3]Regional Brewer Calibration Center for Europe, Izaña Atmospheric Research Center, Tenerife, Spain
[4]National Institute for Aerospace Technology - INTA, Atmospheric Observatory "El Arenosillo", Huelva, Spain
[5]Physikalisch-Meteorologisches Observatorium Davos/World Radiation Center, Davos, Switzerland
[6]Manchester University, Manchester, United Kingdom
[7]Laboratory of Atmospheric Physics, Aristotle University of Thessaloniki, Thessaloniki, Greece
[8] International Ozone Services, Toronto, Canada
[9]Agencia Estatal de Meteorología, Madrid, Spain
[10]National Meteorological Office, Algeria
[11]Danish Meteorological Institute, Copenhagen, Denmark
[12]Kipp & Zonen, Delft, The Netherlands
[13]Scientific and Production Association "Typhoon", Obninsk, Russia
[14]Finnish Meteorological Institute, Sodankyla, Finland

*Correspondence to:* Alberto Redondas (aredondasm@aemet.es)

**Response to Reviewer #2**

**Comment:** The paper "EUBREWNET RBCC-E Huelva 2015 Ozone Brewer Intercomparison" by Redondas et al. describes some of the main findings from an international comparison campaign of Brewer spectrophotometers. After an introduction about the Brewer ozone retrieval algorithm and the calibration transfer techniques, particular attention is given to an empirical parametrisation/correction of stray light applied to the single-monochromator instruments. A short discussion about the "standard lamp correction" to track the radiometric stability of the spectrophotometers is also provided.

In my opinion, the paper potentially raises the following important questions:

1. What is the maximum attainable reproducibility by well-calibrated Brewer spectrophotometers?.

2. What are the most common sources of error/instability in the Brewer measurements? How can they be identified and solved during an intercomparison?.

3. How important is the stray light effect on ozone estimates and what techniques can be used to overcome this issue?.

4. How good is the agreement among reference instruments used to calibrate the Brewer network?.

Therefore, the manuscript, in principle, addresses relevant scientific questions within the scope of ACP. However, I have two main concerns related to the paper:

1. The stray light and the standard lamp corrections should be discussed more properly (cf. Specific comments);

5    2. The manuscript resembles more to a technical report than a scientific article (especially considering that the manuscript has been submitted to ACP).

The previously listed scientific questions (1-4) deserve a deeper discussion, and should be better enhanced (e.g., they should be presented in the introduction, together with a set of bibliographic references, and answered in the main text through quantitative results). Technical details (e.g., determination of the dead time, dispersion function, etc.) that would be suitable for a
10    report should be omitted in the present paper if not relevant to the scientific discussion. A reorganization of the paper, keeping theory and results better apart, would improve readability (cf. Technical corrections).

Once these remarks are properly addressed, the paper can be published in ACP.

**Answer:**

We have reorganised the paper to address these main questions. The stray light and the standard lamp corrections are
15    discussed on the Material and Methods section and the introduction included reference on the main topics.

In this work we focus on the reproducibility of the EUBREWNET network, the precision and accuracy of well maintained triads which serve as references of the Brewer ozone measurements are described in other studies (Fioletov et al., 2005; Stübi et al., 2017; León-Luis et al., 2018) and we think that this is outside the topic of this work.

We add a table of the comparison of the reference instruments during the RBCC-E campaigns and the corresponding cali-
20    bration report. The agreement is generally around +/- 0.5% but with exceptions. In contrast to the RBCC-E travelling reference who is transported by boat/car to Huelva and as hand luggage using two extra-seats of the plane, to Arosa campaigns, the others travelling references (IOS #017 and Kipp&Zonen #158 ) are usually transported by cargo and can have issues during transportation that are reflected in the table. Also, due to the instrumental changes for example #158 has a new PMT and new electronics during Arosa 2014 and the SL do not reflect this change.

25    **Specific comments**

**Comment:** It is rather trivial that the comparison between single- and double-monochromator instruments improves when a stray light empirical correction, obtained from the comparison itself, is applied back to the same set of data. What is not obvious, in my opinion, is that the correction obtained during the intercomparison can be also used to improve accuracy when the Brewer is moved back to the home institution after the campaign. This would be an important conclusion, but some points
30    should be addressed:

**Table 1.** Reference Comparison during RBCC-E campaigns

| Location | year | #017 | #158 | #145 | Report |
|----------|------|------|------|------|--------|
| Arosa | 2008 | -0.6 | | | (Redondas and Rodriguez-Franco, 2008) |
| Huelva | 2009 | -0.6 | 0.8 | -0.1 | (Roozendael et al., 2012) |
| Arosa | 2010 | -0.6 | | | (Roozendael et al., 2013b) |
| Huelva | 2011 | -0.1 | -0.2 | -0.6 | (Roozendael et al., 2013a) |
| Arosa | 2012 | | -0.1 | | (Redondas et al., 2015) |
| Huelva | 2013 | -1.0 | 0.7 | | (Redondas and Rodriguez-Franco, 2015a) |
| Izaña | 2014 | | | -2.2 | (Redondas et al., 2014) |
| Arosa | 2014 | -1.2 | 1.5 | | (Redondas and Rodriguez-Franco, 2015b) |
| Huelva | 2015 | -0.5 | -0.5 | | this work |

1. it should be proved (or discussed) that the correction only depends on the instrumental characteristics and not on the measurement site;

2. stray light should be characterised, during the intercomparison, for the full range of slant ozone values reached during normal operation. This is particularly important for single-monochromator instruments located at high-latitude stations. Can the authors state that the OSC range during the intercomparison is wide enough?

3. the authors affirm that "These parameters, determined in several campaigns, have been found to be stable" (page 6 l. 8). This is a key point: can they show some quantitative data demonstrating that the correction is stable over time?

**Comment:**

**Answer:** We agree with the suggestion and we think we can prove our affirmations. As described in the introduction, one of the regular RBCC-E campaigns were the Nordic campaigns, within these campaigns the FMI MKII Brewer #037 operating at Sodankyla since 1988 was calibrated four times, at Izana in 2009, 2011, and 2015 and at Sodankyla (Finland) in 2011 (Roozendael et al., 2013b, 2014).

From these measurement campaigns, we found

- The stray light correction obtained during the first campaign where applied to the subsequent campaigns obtained a very good agreement, better than 0.5% on the 300-1800 range.

- The correction is valid also when the spectral response of the instrument is changed, we detect a change on the ETC during the last campaign (2013), and the application of the new ETC with the 2009 stray parameters give also very good results.

- The stray light parameters ($k$ and $s$) obtained during different campaigns (Table 2) are in agreement when we consider the confidence interval of the adjustment.

- During the calibrations at Huelva, the measurement schedule is carefully defined to maximize the observations at high airmass. As also described in the campaign conditions Section around 30% of the simultaneous observations are performed

**Table 2.** Summary of the FMI #037 from (Roozendael et al., 2014) calibration constants including the stray light parameters k and s, the intercept F0 and the ETC constant calculated using the standard 1-parameter method (F01P) and the standard lamp R6 ratio reference value (R6ref). R2 is the coefficient of determination for the power-law fitting.

| Campaign | k | k(95%CI) | s | s(95%CI) | $F_0$ | $F_0$(95%CI) | R2 | F01P | R6Ref |
|---|---|---|---|---|---|---|---|---|---|
| Izo2009 | -12 | [-17.58,-6.41] | 4.79 | [3.79,5.78] | 3117 | [3112,3123] | 0.942 | 3115 | 1880 |
| Sdk2011 | -12.66 | [-18.65,-6.67] | 4.56 | [3.88,5.23] | 3104 | [3091,3118] | 0.99 | 3115 | 1880 |
| Izo2011 | -18.29 | [-24.51,-12.07] | 3.97 | [3.19,4.76] | 3106 | [3102,3111] | 0.987 | 3115 | 1880 |
| Izo2013 | -11.37 | [-17.50,-5.25] | 5.54 | [4.42,6.66] | 3119 | [3112,3126] | 0.986 | 3120 | 1870 |

[Figure]

**Figure 1.** Ratio to reference on Single Brewer, the stray light free region is generally up to 800 DU were underestimation start and is determined for each instrument based on the one-parametner/two-parameter calibration agreement

with OSC> 600, 15% of that >900 reaching the 1600 DU. During these campaigns at Izaña, we can get 1800 DU reaching 2000 at Sodankyla.

- Figure 1 shows the ratio of the single brewer participating in the campaign, here we can see that the OSC free region which we use for calibration start for some instruments at 600DU and is almost evident at 1000DU, utilizing observations up to 1600

5   DU we have enough measurements to determine the stray light constants.

**Comment:** The section about the SL correction (page 9) is quite inconclusive. It is demonstrated that the SL correction does not improve the accuracy for some Brewers (while it does for others) and that the only way to verify it is an intercomparison against a reference instrument. In that case, how should the Brewer data be reprocessed from one intercomparison to the

next? My concern is not to spread the idea that the Brewer data quality is aleatory and that the community does not know how to reprocess the data for improving their quality.

**Answer:** We try to rephrase this but this is exactly the fact, the Brewer calibration is tracked by the SL measurements, without any external comparison we are not able to determine if the internal lamp properly tracks the changes of the spectral response of the instrument or not. This is the main reason why a regular calibration of the instrument is needed (every two years) and why the final data in EUBREWNET (Level 2.0) is achieved after the calibration. Of course stable instruments have also stable SL record and maintain their calibration and no reprocessing is required, for our experience small changes are tracked very well with the standard lamp but this does not always happen with huge changes due mostly to major issues in the instrument.

**Technical corrections**

**Comment:** page 1 l. 4-6: omit the reference to UV and QASUME in the abstract if the UV results are not discussed in the text;
**Answer:** removed

**Comment:** page 1 l. 4: "Twenty-one". It could be useful to mention already in the abstract how many single- and double-monochromator instruments have been studied;
**Answer:** Added the number of doubles and singles, also this information is added to the table. The proportion of single and double Brewer in the campaign is approximately the same as the network

**Comment:** page 1 l. 7: at the first occurrence, use "spectral stray light" instead of only "stray light", to distinguish it from other sources of stray light (e.g., multiple scatter stray light n the field of view);
**Answer:**
Added a description of the stray light as was also requested by Referee #1

**Comment:** page 1 l. 9-10: omit 76% and 50% percentages (16/21 is easy to calculate, and 10/21 is 47%, not 50%);
**Answer:** done. For the statistics we do not count the instrument #151 as it is not operative and is not providing observations.

**Comment:** page 1 l. 10: state the air mass range relative to the 1% and 0.5% thresholds;
**Answer:** included the ozone slant column rather than the airmass range, as the stray light effect depend on this.

**Comment:** page 1 l. 11: enhance the outcomes of the paper, e.g. why / to whom those findings are important?
**Answer:** added, see general comments.

**Comment:** page 2, Sect. 1: I would expect a more sound introduction, focusing on scientific issues (and related literature, e.g. on the stray light effect, etc.), rather than a summary of the previous campaigns. Also, since the previous technical reports

are mentioned, it should be specified what the novelty of the present study is;

**Answer:** Each RBCC-E campaign advances our understanding of the Brewer calibrations in general and specifically what areas still require attention. In this 2015 campaign we have introduced a formal approach to characterization of the internal instrumental stray-light , the filter attenuation correction and the algorithm for correcting for its effects on the TOC calculations, and these improvements were introduced on the EUBREWNET calculation.

**Comment:** page 2 l. 17: "Figure" instead of "Fig." at the be- ginning of a sentence

**Answer:** done .

**Comment:** page 3 l. 2: table 2 is cited before table 1, please reverse the order of tables;

**Answer:** done

**Comment:** page 3 l. 3: the measurement site should be better described, since the local characteristics of the measurement site impact of the results of the campaign;

**Answer:** A description of the site and the conditions of the intercomparison has been added.

**Comment:** page 3 l. 4-5: since UV radiation is out of the scope of this paper, omit the references to QASUME and WRC-UV;

**Answer:** The UV radiation is not covered by the paper but is an important aspect of the campaign. Moreover some of the aspect of the calibration are cross related like the wavelength calibration.

**Comment:** page 3 l. 8-21: those paragraphs have nothing to do with Sect. 1.1, entitled "The X RBCC-E campaign". Omit them and reference the EUBREWNET paper (https://www.atmos-chem-phys-discuss.net/acp-2017-1207/) in your Intro-duction;

**Answer:** done

**Comment:** page 4 Eq. 1: if alpha is defined as in Eq. 3, Eq. 1 should read (ETC-F)/(alpha*mu) instead of (F-ETC)/(alpha*mu). Otherwise, if alpha is defined negative (as in the Brewer literature), Eq. 3 should have a minus sign;

**Answer:** corrected

**Comment:** page 4 l. 8: define what "double ratios" are, for the unexperienced reader. "corrected for the Rayleigh effects" -> "to which the effect of Rayleigh scattering has been subtracted (Eq. 2)";

**Answer:** corrected in text.

**Comment:** page 4 l. 16: "verify" -> "satisfy";

**Answer:** done

**Comment:** page 4 Eq. 5: Eq. 6 is explained in the following text, please explain Eq. 5 as well;

**Answer:** added an explanation and reference

**Comment:** page 5 l. 1: "calibration" -> do you rather mean "characterization"?

> **Answer:** corrected

**Comment:** page 5 l. 2: "filter attenuation" -> has any filter already been mentioned in the text?

> **Answer:** A short description of the instrument is added.

5  **Comment:** page 5 l. 3: "The wavelength calibration allows to..." -> "An accurate determination of the operational wavelengths is needed to". "are" -> "is";

> **Answer:** The paragraph has been modified following your suggestion

**Comment:** page 5 l. 6: "Finally, the ETC transfer is performed by comparison..." -> "Finally, the ETC must be determined by transfer from a reference Brewer (Sect. 2.1) or, in the case of the reference instruments, by the Langley method" (include

10     bibliographic references about the Langley method);

> **Answer:** The sentence has been rephrased and some references to the Langley method were added.

**Comment:** page 5 l. 8-9: "changes ... wavelength calibration will affect the final ETC and changes in the wavelength calibration will affect also the final ETC" -> is there any difference between the two sentences?

> **Answer:** corrected

15  **Comment:** page 5 l. 18: "calibration" -> "radiometric";

> **Answer:** The word "radiometric" suggests an absolute calibration of the instrument. However, the Brewers present a relative calibration, so we think that it is more correct to use the word "calibration" for this article.

**Comment:** page 5 l. 20: "for simultaneous measurements BY BOTH INSTRUMENTS";

> **Answer:** corrected

20  **Comment:** page 5 Eq. 7: explain that "i" does not refer to wavelength, as in Eqs. 2-6, but to the sample (better: use a different subscript). Correct the sign "-" to "+" according to Eq. 2 (cf. my previous comment);

> **Answer:** corrected

**Comment:** page 5 l. 27: define the "stray-light free region" from a quantitative point of view;

> **Answer:** A detailed definition of the stray-light free region is defined, from a quantitative point of view this could be

25     defined as the region where the underestimation of the ozone is less than 0.3% but it is difficult to address because this difference will also depend on the calibration constants of the instrument. To avoid this the agreement of one point calibration with the two point calibration is used to determine the stray light free region for every instrument, which range from 900 DU to 600 DU.

**Comment:** page 6 Fig. 3: Figs. 3 and 4 are very similar. Consider keeping only one of them. Add units of ozone slant path (cm STP);

**Answer:** The figure 3 is the standard operating procedure where as the Figure 4 shows the stray light calculation, as the data are the same we try to use only Figure 4 but then the standard operating procedure is not clear, so we think that is worthwhile to maintain the two, figures.

**Comment:** page 6 Eq. 8: the "F0" notation of Eq. 4 may be confused with "Fo" of Eq. 8. Consider using a different notation;

**Answer:** corrected

**Comment:** page 6 Eq. 8 and 9: as far as I understand, the $k(X*mu)^2$ term can be attributed to either the F's or to the ETC's, but not to both of them at the same time. In this case, the text should be clearer;

**Answer:** Mathematically it can be considered to both terms, it is a correction to the measured counts but as is determined with the ETC, and the correction is similar to to the Standard Lamp correction. This is corrected in the text.

**Comment:** page 6 line 7: "stray-light free OSC region" -> this expression is misleading, since the $k(X*mu)^s$ cannot be zero for mu>=1 (or reformulate the correction term to be zero for mu=1). Also, could the ETC be directly retrieved from the fit, together with k and s?

**Answer:**

The non-lineal model can also retrieve ETC and the ozone absorption coefficient from the comparison with the reference (ETCs,O3ABSs,K,S). In this case the correction is zero for m=1. Indeed we perform these calculations and use the comparison witht the ETCo, and as a check of the calibration. The effect on the stray light parameters are small.

Using the ETC (and the ozone absorption coefficient) from the one point calibration, different from the calculated of the non-linear model produces an generally small offset. In most of the cases the difference is very small,on the worst cases produce an offset 0.3% (Figure 2) on low OSC due to the difference between the ETC/absorption coefficient determined by the operational method and that derived from the fit.

**Comment:** page 7 Eq. 10: since k is negative, if we want the ozone to increase at the next interaction, there must be a "-" sign, unless alpha is defined negative (Eq. 3). Please, clarify;

**Answer:** The sign was incorrect ,and corrected now.

**Comment:** page 7 line 5: "2000 DU" -> to what air mass does it correspond, in El Arenosillo? Is the calculation of the air mass reliable at this SZA? Notice that the stray light correction presented in the paper depends on the product X*mu: if X is low during the determination of the correction, mu must be large (but still accurate!) to cover a sufficiently wide range of OSC, since, at high latitudes, the same OSC can be reached at lower mu's;

**Answer:** In most of the Brewer spectrometers there is a physical limitation of the observation to 80º, which corresponds to an air-mass around 5 and OSC of 1600 DU during the campaign. The measurement schedule has been adapted to have as much as possible measurements at short angles.

[Figure]

**Figure 2.** Stray Light correction when the ETC is also derived of the fit, there is an ofset at low OSC due the different ETC but the effect on the stray light constants are small.

**Comment:** page 7 "3.1 Reference Calibration" -> or "Agreement between reference instruments"?

    **Answer:** Changed and added reference.

**Comment:** page 8 l. 1-6: the comparison of the reference instruments is an interesting topic, and the RBCC campaigns a very unique chance to investigate it. However, the authors should discuss it with more details. E.g., how are the ECCC Toronto Triad and #158 calibrated? The procedure is explained for the IZO triad, but not for other reference instruments;

    **Answer:** This specific topic is discussed in the work of León-Luis et al. (2018) where different triads are compared, we add in the text a reference for this work

**Comment:** page 8 l. 5-6: why two different thresholds (900 DU and 600 DU)?

    **Answer:** The #017 is a special case of an instrument with strong stray light, the statistics are calculated up to 900DU, but the underestimation of the ozone is noticeable after 600 DU.

**Comment:** page 8 l.8 to page 9 l. 7: this basically theoretical part should be described in the "Methods" section, not in the "Results";

    **Answer:** moved.

**Comment:** page 9 l. 13: Brewer #165 does not exist in the list of the instruments;

    **Answer:** corrected.

**Comment:** page 10 l. 3: "Brewer #151 can not be considered an operational instrument" -> what does this sentence mean?

**Answer:** Not all participating instruments are operative, as we include in the text , the non-operational instruments are the instruments that can't provide reliable data and have to be fixed during the campaign. Like the case of #151 that was fixed during the campaign.

5  **Comment:** page 10 l. 4: "ago" -> "before";

**Answer:** The word was changed in the text.

**Comment:** page 10 l. 12 and 14: "more than 5 units during this period" repeated twice;

**Answer:** Yes, we have deleted this duplicate phrase.

**Comment:** page 11 Fig. 9: ordering the x-axis by serial number would improve readability of the chart;

10  **Answer:** Done as suggested.

**Comment:** page 12 Fig. 10: both the bias and the spread for Brewers 158 and 228 is large (in the boxplot). Is there any connection between bias and spread (e.g., instability of the lamps...)?

**Answer:** There is no relation between bias, which gives us an idea of the instrument change between calibration, and the spread during the campaign periods.

15  **Comment:** page 12 l. 1-5: move these lines to a "Methods" section;

**Answer:** These sentences were moved following the suggestion of the referee.

**Comment:** page 13 l. 4: if the discussed results were representative of the overall Brewer network, this would mean that about 25% of the instruments are "out of spec" after two years. This is a strong statement: could the author elaborate on this?

**Answer:** This is what the data suggest, and this is the reason why a two-year calibration is advised. - The overall +/- 1%

20  agreement of the Brewer network as is assumed by the community has been established by the triads of well maintained instrument, but is not done using network instruments. The result is 25 % of the instruments are outside 1% but none of the instruments were outside 1.5%.

**Comment:** page 13 l. 7: "celebrated" -> "taking place"?

**Answer:** corrected

25  **Comment:** page 14: can the authors draw more definitive and general conclusions from the study?

**Answer:** We have added the conclusions of the study adding the stray light discussion.

**Comment:** page 15 Fig. 13: could the variations over time be explained by the variability of the participating instruments? This should be mentioned in the text. Caption: what does "no operating" mean?

**Answer:**

Yes In effect the variability can be due that the participating instruments are not always the same and can reflect the variability some Brewers only participated in 1-2 campaigns their results may affect overall agreement spread.

**Notes**

**References**

Fioletov, V. E., Kerr, J., McElroy, C., Wardle, D., Savastiouk, V., and Grajnar, T.: The Brewer reference triad, Geophysical Research Letters, 32, https://doi.org/10.1029/2005GL024244, http://doi.wiley.com/10.1029/2005GL024244, 2005.

León-Luis, S. F., Redondas, A., Carreño, V., López-Solano, J., Berjón, A., Hernández-Cruz, B., and Santana-Díaz, D.: Stability of the Regional Brewer Calibration Center for Europe Triad during the period 2005-2016, Atmos. Meas. Tech. Discuss., 2018, 1–20, https://doi.org/10.5194/amt-2017-460, https://www.atmos-meas-tech-discuss.net/amt-2017-460/, 2018.

Redondas, A. and Rodriguez-Franco, J.: Eighth Intercomparison Campaign of the Regional Brewer Calibration Center Europe (RBCC-E), no. 223 in GAW Report, World Meteorological Organization, http://library.wmo.int/pmb_ged/gaw_223_en.pdf, 2015a.

Redondas, A. and Rodriguez-Franco, J.: Ninth Intercomparison Campaign of the Regional Brewer Calibration Center Europe (RBCC-E), no. 224 in GAW Report, World Meteorological Organization, http://library.wmo.int/pmb_ged/gaw_224_en.pdf, 2015b.

Redondas, A. and Rodriguez-Franco, J. J.: AROSA 2008 REPORTS - WikiNesia, Calibration Report, http://www.iberonesia.net/index.php/AROSA_2008_REPORTS, 2008.

Redondas, A., Rodriguez, J., Carreno, V., and Sierra, M.: RBCC-E 2014 langley campaign the triad before Arosa/Davos campaing, https://repositorio.aemet.es/bitstream/20.500.11765/2599/1/Alberto%20TriadArosaDavos.pdf, 2014.

Redondas, A., Rodríguez-Franco, J., Gröbner, J., Köhler, U., and Stuebi, R.: Seventh Intercomparison Campaign of the Regional Brewer Calibration Center Europe (RBCC-E), no. 216 in WMO/GAW Reports, World Meteorological Organization, http://www.wmo.int/pages/prog/arep/gaw/documents/Final_GAW_216.pdf, 2015.

Roozendael, M. V., Köhler, U., Pappalardo, G., Kyrö, E., Redondas, A., Wittrock, F., Amodeo, A., and Pinardi, G.: CEOS Intercalibration of Ground-Based Spectrometers and Lidars: Second Progress Report, Tech. rep., European Space Agency, http://repositorio.aemet.es/handle/20.500.11765/8889, 2012.

Roozendael, M. V., Köhler, U., Pappalardo, G., Kyro, E., Redondas, A., Wittrock, F., Amodeo, A., and Pinardi, G.: CEOS Intercalibration of Ground-Based Spectrometers and Lidars: Final report, Tech. rep., European Space Agency, http://repositorio.aemet.es/handle/20.500.11765/8886, 2013a.

Roozendael, M. V., Köhler, U., Pappalardo, G., Kyrö, E., Redondas, A., Wittrock, F., Amodeo, A., and Pinardi, G.: CEOS Intercalibration of Ground-Based Spectrometers and Lidars: Final report, Tech. rep., European Space Agency, http://repositorio.aemet.es/handle/20.500.11765/8886, 2013b.

Roozendael, M. V., Kyrö, E., Karppinen, T., Redondas, A., Cede, A., and Hendrick, F.: CEOS Intercalibration of Ground-Based Spectrometers and Lidars: Contract Change Notice 2012-2013: Final Report, Tech. rep., European Space Agency, http://repositorio.aemet.es/handle/20.500.11765/8909, 2014.

Stübi, R., Schill, H., Klausen, J., Vuilleumier, L., and Ruffieux, D.: Reproducibility of total ozone column monitoring by the Arosa Brewer spectrophotometer triad: AROSA BREWER TRIAD, Journal of Geophysical Research: Atmospheres, 122, 4735–4745, https://doi.org/10.1002/2016JD025735, http://doi.wiley.com/10.1002/2016JD025735, 2017.

---

## Author Response (AR1)

**EUBREWNET RBCC-E Huelva 2015 Ozone Brewer Intercomparison**

Alberto Redondas[1,2], Virgilio Carreño[1,2], Sergio F. León-Luis[1,3], Bentorey Hernández-Cruz[2,3], Javier López-Solano[1,2,3], Juan J. Rodriguez-Franco[1,3], José M. Vilaplana[4], Julian Gröbner[5], John Rimmer[6], Alkiviadis F. Bais[7], Volodya Savastiouk[8], Juan R. Moreta[9], Lamine Boulkelia[10], Nis Jepsen[11], Keith M. Wilson[12], Vadim Shirotov[13], and Tomi Karppinen[14]

[1]Izaña Atmospheric Research Center, Agencia Estatal de Meteorología, Tenerife, Spain
[2]Departamento de Ingeniería Industrial, Universidad de La Laguna, Tenerife, Spain
[3]Regional Brewer Calibration Center for Europe, Izaña Atmospheric Research Center, Tenerife, Spain
[4]National Institute for Aerospace Technology - INTA, Atmospheric Observatory "El Arenosillo", Huelva, Spain
[5]Physikalisch-Meteorologisches Observatorium Davos/World Radiation Center, Davos, Switzerland
[6]Manchester University, Manchester, United Kingdom
[7]Laboratory of Atmospheric Physics, Aristotle University of Thessaloniki, Thessaloniki, Greece
[8] International Ozone Services, Toronto, Canada
[9]Agencia Estatal de Meteorología, Madrid, Spain
[10]National Meteorological Office, Algeria
[11]Danish Meteorological Institute, Copenhagen, Denmark
[12]Kipp & Zonen, Delft, The Netherlands
[13]Scientific and Production Association "Typhoon", Obninsk, Russia
[14]Finnish Meteorological Institute, Sodankyla, Finland

*Correspondence to:* Alberto Redondas (aredondasm@aemet.es)

**Abstract.** From May 25[th] to June 5[th] 2015, the [..[1] ]10th Regional intercomparison campaign of the Brewer Calibration Center - Europe (RBCC-E) was held at El Arenosillo atmospheric sounding station of the Instituto Nacional de Técnica Aeroespacial (INTA). This campaign was [..[2] ]jointly conduced by COST Action ES1207 EUBREWNET and the Area of Instrumentation and Atmospheric Research of INTA. Twenty one [..[3] ]Brewers, 11 singles and 10 double monochromator instruments from eleven countries participated and [..[4] ]were calibrated for total column ozone and solar UV irradiance[..[5] ]

. Every RBCC-E campaign advances our understanding of the Brewer calibrations in general and specifically what areas still require attention. In this 2015 campaign we have introduced a formal approach to characterization of the internal instrumental stray-light, the filter non-linearity and the algorithm for correcting for its effects on the TOC calculations. This work shows a general overview of the ozone comparison[..[6] ], the evaluation of the correction of the spectral stray light effect for the single-monochromator Brewer spectrophotometer, derived from the comparison with a reference double-
* * *
[1]removed: X

[2]removed: a joint effort of

[3]removed: Brewer

[4]removed: their

[5]removed: calibrations were performed, in the latter case using the traveling reference standard QASUME instrument of the World Radiation Center for UV (WRC-UV).

[6]removed: focused on the

[revised manuscript text omitted]

[32]removed: verify:

[33]removed: in local

[34]removed: .

[35]removed: We can divide the calibration in instrumental, wavelength, and ETC transfer steps:

[36]removed: measured counts

[37]removed: ((Fountoulakis et al., 2016))

[38]removed: and filter attenuation .

[39]removed: allows to determine

[40]removed: coefficient. The

[41]removed: are used to obtain the particular wavelength for the instrument and the

its slit, or instrumental[..[42] ][..[43] ], function (Gröbner et al., 1998; Redondas et al., 2014a). An extra-terrestrial (calibration) constant is determined by the Langley method or by comparison with a reference instrument.

It is important to note that TOC It is then finally determined using ratios of measurements so there is no transfer of the radiometric scale. During the campaigns the transfer of the calibration to a network instrument is achieved by operating side by side with the reference [..[44] ]Brewer. Once we have collected enough near-simultaneous direct sun ozone measurements, we calculate the new extra-terrestrial constant after imposing the condition that the measured ozone will be the same for simultaneous measurements for both instruments. In terms of Eq. 1, this leads to the following condition:

$$ETC_j = F_j + X_j^{reference}\alpha\mu_j \tag{7}$$

In the Langley determination of the ETC we assume that the ozone, as well other interfering absorbents, is constant through half of a day [see Eq. (7)] thus the regression of the double ratios $F_j$ against the airmass will be a line with slope $\alpha X$ and intercept $ETC$ (Komhyr et al., 1989) . These assumptions are only achievable in clear environments and high altitude subtropical stations like Izaña and Mauna Loa. The Langley calibration of the RBCC-E instruments are described in detail in (Redondas et al., 2014b; León-Luis et al., 2018) and are outside the focus of this work.

For a correctly characterised network instrument, the determined ETC values show a Gaussian distribution and the mean value is used as the instrument's extra-terrestrial constant. One exception to this rule is the single monochromator Brewer models (MK-II and MK-IV) which are affected by stray light (Karppinen et al., 2015). In this case, the ETC distribution shows a tail at the lower ETC values for high Ozone Slant Column (OSC, the product of the total ozone content by the airmass). As we discuss in detail on the next section for this type of Brewer, only the stray-light-free region is used to determine the ETC, which generally ranges from 300 to 800 DU in the OSC, depending on the instrument.

The network Brewers were calibrated using the one parameter ETC transfer method, i.e., the ozone absorption coefficient was derived from the wavelength calibration (dispersion test) and only the ozone ETC constant was transferred from the reference instrument. The so-called "two parameters calibration method" (Staehelin et al., 2003), where the ozone absorption coefficient is also calculated from the reference, is also calculated and used as a quality indicator.

The calibration is an iterative process[..[45] ], changes during the instrumental and/or wavelength calibration will affect the final ETC[..[46] ], some instrumental characteristics are revealed during the ETC transfer by the comparison with the reference which requires full reprocessing of the calibration. For this reason the calibration campaigns are scheduled in three different periods:
* * *
[42]removed: function, of each spectrophotometer, which differs slightly from instrumentto instrument (Redondas et al., 2014a).

[43]removed: Finally, the ETC transfer is performed by comparison

[44]removed: or, in the case of the reference instruments , by the Langley method.

[45]removed: –

[46]removed: and changes in the wavelength calibration will affect also to the final ETC

[revised manuscript text omitted]

* * *
[69]removed: The instruments are working during this period with their home calibration and the ozone is calculated using these calibration constants. The Standard Lamp (SL) test is an ozone measurement using the internal halogen lamp as a source. This test is performed routinely to track the spectral response of the instrument and, therefore, the ozone calibration. A reference value for the SL, the so-called R6 ratio, is provided as part of the calibration of the instrument. The ozone is routinely corrected assuming that deviations of the R6 value from the reference value are the same as the changes in the ETC Extraterrestrial constant. This then described by the Standard Lamp correction:

[revised manuscript text omitted]

- 10 Brewer spectrophotometers ($\sim$50%) were within the ±0.5% range, i.e., show a perfect agreement.

- The max average error was 1.5% for operational Brewer instruments within stray-light free conditions (OSC $<$ 700 [..$^{215}$ ]DU).
* * *
$^{213}$removed: To summarize
$^{214}$removed: X
$^{215}$removed: DU

This results are in agreement with the RBCC-E campaigns celebrated in Huelva and Arosa from 2009 to 2015 (Figure 14), in this period 85 spectrometers [..216 ]have been calibrated: 59 (69%) [..217 ]show an agreement better than 1%, 32 (38%) within 0.5% and 7 (8%) [..218 ]show a discrepancy greater than 2%.

After the new calibration was issued at the end of the X RBCC-E campaign,

- All participating Brewer spectrophotometers were within the ±0.5% agreement range.

[revised manuscript text omitted]

---

## Author Response (AR2)

**Answer to the referee**

We want to thank to the referee for the comments, to ask the main question: the problem of extrapolating the stray light correction. We have perform an statistical analysis of the prediction bounds of the fit that will be further included in the error budget analysis of the TOC of the Brewer. To probe the reliability of the extrapolation we have used observations from the Nordic campaign with the Brewer #037 that was also used by Karppinen 2005 work, this allows us to have a more complete range of OSC and the comparison with the Karppinen model . Unfortunately the Brewer #037 do not participate in the campaign and we do not consider appropriate to include the plots on this paper.

**p. 1 l. 6: can you explain better what areas still require attention? Reformulate the sentence to be grammatically consistent**

*Every RBCC-E campaign advances our understanding of the Brewer calibrations in general and specifically what areas still require attention. In this 2015 campaign we have introduced a formal approach to characterization of the internal instrumental stray-light, the filter non-linearity and the algorithm for correcting for its effects on the TOC calculations.*

Rephrase, the sentences have been moved to the introduction:

Before the establishment of the RBCC-E, the Brewer spectrophotometer calibrations were referenced to the Brewer World Calibration Centre hosted by Environment Canada (EC). However, most of the Brewer instruments were, and still are, calibrated by private companies, in the main by International Ozone Services (IOS) and to a lesser extent by Kipp and Zonen bv (Staehelin, 2010). The RBCC-E calibration adapts the methodologies and tools developed by EC and IOS, but also investigates and improves particular issues. The foci of the first campaigns were related to the instrument characterization and the ozone absorption calculation (Redondas and Rodriguez-Franco, 2012) whereas in this campaign the focus was on the stray light correction and the investigation of the error due to non-linear filter attenuation.

Staehelin, J.: TOTAL OZONE MONITORING BY GROUNDBASED INSTRUMENTS AS PART OF GAW, in TECO-2010 - WMO Technical Conference on Meteorological and Environmental Instruments and Methods of Observation, p. 12, WMO-GAW, Helsinki. [online] Available from: https://www.wmo.int/pages/prog/www/IMOP/publications/IOM-104_TECO-2010/1_3_Stahelin_Switzerland.pdf, 2010.

**p. 1 l. 7: could you add a chart of the filter non-linearity effect in the paper? It is mentioned in the abstract as an important point, however only a text description is provided in the paper;**

Figure 6 were added:

[Figure]

**Figure 6.** BoxPlot of the percentual differences with respect to the reference grouped by filter for Brewer #186, in blue without correction, and in black after applying the correction to filters 3 and 4 (upper panel). On the lower panel percentage differences with respect to the reference grouped by filter, without correction (solid dots), and after the application of the correction to filters 3 and 4 (open circles). Colors indicate the number of the filter; see the legend.

**p. 1 l. 8: "its effects" refers to stray-light or non-linearity? "its" or "their"?**

It refers to both the stray-light and non-linearity.

**p. 3 l. 11: "also"..."also" - remove one recurrence;**

Corrected

**p. 3 l. 15: "the" results;**

Corrected

**p. 4 l. 8: "measurements" -> "measurement";**

Corrected

**p. 4 l. 14: add comma ("filter, a");**

Corrected

**p. 4 l. 18: "because of using a unique grating". I would say "monochromator", not grating, which is only one component of the monochromator;**

Corrected:

```
However, both instruments are affected by stray light, mainly because both
use a single  monochromator. In contrast, the MKIII model is a double
monochromator which provides enough stray light rejection to work in the
first diffraction order for UV and ozone measurements.
```

**p. 5 l. 2: Eq. 1 was not corrected. It should read (ETC-F)/alpha*mu;**

Corrected:

```
\begin{equation}
      \label{eq:ozone}
      X = \frac{{ ETC -F }}{\alpha \mu }\
\end{equation}
```

**p. 5 l. 4: "ozone differential absorption coefficient";**

```
where $F$ is a linear combination of the algorithm of the measured spectral
direct irradiances (also called double ratios) corrected for Rayleigh molecular
scattering, $\alpha$ is the ozone differential absorption coefficient, $\mu$ is
the ozone air mass factor, and $ETC$ is the extra-terrestrial constant. The $F$,
$\alpha$ and $ETC$ parameters are weighted functions at the operational
wavelengths:
```

**p. 5 l. 9-18: those lines are confusing. The authors talk about weights (SO2 and aerosol), then about the wavelength choice (and the dispersion test is only described at p. 6 l. 4-6 - by the way, F0 at line 12 is not defined), then again about the weights**

**(wavelength-independent factors and aerosol). Please re-order this paragraph;**

**We agree with referee the paragraph has been changed:**

```
 The four longer wavelengths (310.1, 313.5, 316.8  and 320.1 \unit{nm}) are used
for the ozone calculation with weightings$w=[1,-0.5,-2.2, 1.7]$, respectively.
These wavelengths have been selected near stationary points in the ozone absorption
spectrum and are thus optimized to minimize the influence of small wavelength
shifts (i.e. $\delta F // \delta \lambda =0$). The weightings were determined to
suppress the influence of $SO_2$ and aerosol. Moreover, as $\lambda_i$ and $w_i$
satisfy the conditions defined by Eqs. (\ref{eq:sum_w}) and (\ref{eq:sum_wl}), the
measurement independent of wavelength-independent parameters such as the absolute
calibration. Also, it largely eliminates absorption processes which depend, to
first approximation, linearly on the wavelength,such as the contribution from
aerosols  \citep{kerr_Brewer_2010}.
```

**p. 6 l. 6: "... Langley method (Redondas et al., 2014b; Leon-Luis et al., 2018), in case of reference Brewers, or by comparison with a reference instrument in the case of the other Brewers of the network". Moreover, I would eliminate l. 14-18, since the Langley technique is not a topic of this paper;**

We agree; the paragraph has been changed.

**p. 6 l. 26: the ozone coefficient is not derived from the dispersion test itself, but from calculations based on spectroscopic datasets and the wavelengths chosen from the dispersion test;**

We agree, the text has been rewritten as

```
The network Brewers were calibrated using the one parameter ETC transfer
method, i.e., the ozone differential absorption coefficient was derived from
the calculations of the wavelength calibration (the so-called "dispersion
test", see Redondas et al., 2018-01-31) applied to the spectroscopic set
of ozone cross sections, and only the ozone ETC constant was
transferred from the reference instrument. The so-called "two parameters
calibration method" (Staehelin et al., 2003), where both the ozone
absorption coefficient and the ETC are calculated from the reference, is also
obtained and used as a quality indicator
```

**p. 6 l. 29: "iterative process, i.e. changes..." and reformulate the sentence, which is too long. Also, could you explain what are "some instrumental characteristics revealed during the ETC transfer" that require full reprocessing?**

The calibration is an iterative process because changes during the instrumental and/or wavelength calibration will affect the final ETC. Some instrumental characteristics which have been improperly accounted for, such as the non-linearity of the filters and the dead-time mismatch (Rodriguez-Franco et al., 2014), are revealed by the comparison with the reference during the ETC transfer. A change in the instrumental constants then requires a full reprocessing of the calibration. For this reason the calibration campaigns are scheduled in three different periods:

**p. 7 l. 5: "The calibrations are maintained between transfer" -> do you mean that changes in the spectral sensitivity are monitored and tracked using the internal quartz halogen lamp? Please, reformulate;**

We agree with the referee, the text has been rewritten as:

The changes in the spectral sensitivity of the instruments are tracked between calibration transfers using the measurements of the internal quartz halogen lamp, the so-called Standard Lamp (SL) test. A value, corresponding to a fictitious column density and often called R6, R5 or F-ratio (depending on whether the ozone, sulphur dioxide or nitrogen dioxide processing algorithm is applied), is obtained after processing.

**p. 7 l. 19-22: I wouldn't go into details about the FOV straylight, since this is not addressed in the present paper. Just remove the lines after the bibliographic reference to Josefsson;**

Removed

**p. 7 l. 24: "such as" -> "so that"?**

Corrected

**p. 8 Fig. 3: the label of the y-axis is not understandable by the inexperienced reader. Caption: "0.6 cm". "Compared with the 1 point calibration": where are the results?**

Corrected:

The reference to two point calibration is removed.

**p. 8 l. 6: the mathematical notation is inconsistent throughout the paper. In the previous text, F was the linear combination of the measured counts, now it represent the true counts and Fm the combination of the measured counts. Please, choose a notation and keep it throughout the paper. Also, F are NOT the true counts, but the linear combination**

**of the true counts. Finally, mention that k<0;**

agree

where $F$ are the true weighted ratios (Eq \ref{eq:double_ratios}), $F_m$, the measured ones, and the k parameter are negative. This is equivalent to correcting the extraterrestrial constant.

**p. 8 l. 8: "This is equivalent to correcting the extraterrestrial constant, i.e.";**
Corrected

**p. 8 l. 11: "... ozone calibration, as explained in the text below.";**

where $F$ are the true weighted ratios (Eq \ref{eq:double_ratios}) and $F_m$, the measured ones. This is equivalent to correcting the extraterrestrial constant.

\begin{equation}

ETC_i= ETC_0 + k {(X \mu)}^s

\end{equation}

where $ETC_0$ is the ETC for the OSC region which is free of stray light, and $k$ and $s$ are retrieved from the reference comparison (Figure~\ref{fig:stray_det}). These parameters, determined in several campaigns, have been found to be stable and independent of the ozone calibration.

**p. 9 l. 14: "OSC free region" -> "straylight-free region". Additionally, what does "starts" mean here?**

Corrected to "end"

**p. 9 l. 15: can you quantify "enough"?-**

-we need observations to derive the k and s parameters, observations up to 1600 seem enough to properly determine the parameters- (See detailed explanation on p10.10 section)

**p. 10 l. 8: if you mention the "model from Karppinen", please first define it;**

We have added a reference to the model

**p. 10 l. 8: can you quantify "significant"? ,**

During the campaign there were 97 simultaneous observations with OSC>1600, but they are not equally distributed between all instruments, so we have decided to remove the comment as it is not relevant.

**p. 10 l. 10: how can the authors use the empirical correction as the reference term for the comparison at those OSCs? This is a point that I raised previously: first the authors say that "getting observations to 1600 DU is enough", but then they extrapolate the correction for comparison up to 2000 DU. Taking as an example the k and s values from Fig. 4, the straylight correction doubles from 1600 to 2000 DU (-232 to -466), so extrapolating above 1600 DU is very dangerous. Please, clarify this fundamental point;**

[Figure]

Figure 1: Figure 4 of the paper extended to osc 2 cm and prediction interval added

Using data up to 1600 DU we can provide a good determination (r-square of the fit better than 0.99) of the non-linear parameters k and s. Once these parameters are determined, the correction can then be applied to any range. We have extended figure 4 with the prediction interval up to 2000 DU, as provided by Matlab's fit function. The correction follows a power law but is well reproduced by the model and the maximum values of the 95% confidence limits are small compared to the correction.

[Figure]

Figure 2: Stray light correction bounds

In the case of #117 the 95% confidence limits reach the +/-1% at 2500 DU, and its value is approx. 0.5% at 2000 DU.

[Figure]

Figure 3: Figure: The same observations as Figure 4 of the paper, but converted to ozone correction, plotted together with the 95% confidence limits. Points are real observations during the campaign and the lines extend up to 3000 DU.

As the referee pointed out, the error bounds increase exponentially with the extrapolation. Using the data of the Brewer #037 at Izaña Nordic campaign, the error at 2000 DU increases from +/-1% if we use observation up to 1600 to 5% if we use only observations up to 1400. The error on the extrapolation will also depend on the particular instrument, on the campaign with observations up to 1600DU the error bounds at 2000 DU for all the

instructions are lower than +/- 2 %.

[Figure]

Figure 4: Stray light correction error bounds for the single brewer at the campaign

We have also compared for Brewer #037 at Izaña  the stray light correction error bounds predicted by the model with the correction finally obtained with  fits with different extrapolations from 1400 to 1700 DU . The mean of the correction applied are very similar in all cases  with less than 0.4% at the highest measured OSC (~1800 DU)

[Figure]

Figure 5: Ration to the reference of the corrected stray light data using   for the fit observations up to osc 1.2,1.4,1.6 and 1.8 cm

[Figure]

Figure 6: % Differences of the correction of the different models to the fit that use all the observations

**Rephrased**

Figure \ref{fig:stray_osc} shows the ratio of the TOC calculated by single Brewer participating in the campaign with respect to the reference instrument, where we can see that the stray-light free region which we use for calibration ends for some instruments at 600 \unit{DU} and is almost evident at 1000\unit{DU}.

The error due to the extrapolation can be estimated from the fit, these error bounds will depend on every instrument and strongly on the extrapolation (figure stray det). During the Nordic campaign the error bounds for Brewer #037 at 2000 DU increases from +/- 0.25%, if we use observations up to 1600 DU, to 1.7% if we use only observations up to 1200 DU. During this campaign with observations up to 1600 DU, the error bounds at 2000 DU are lower than 2% for all the instruments. This results are consistent with the results obtained from the model of Karpinen that shows an error of 1.29 % on 1900-2000 DU interval ( Table 2 of karpinen). To help the determination of these parameters the measurement schedule is carefully defined to maximize the observations at high OSC, where around 30\% of the simultaneous observations are performed with OSC> 600 and 15\% are obtained with OSC >900 DU.

**p. 11 l. 2: who is the subject of "has"? Please correct;**

Differences up to 20 ETC units (around 4% in ozone) have been observed during the campaign.

**p. 11 l. 6: F0 was never defined before;**

Changed by Fm measured in consistence with the stray light correction:

```
where $F$ are the true weighted ratios (Eq \ref{eq:double_ratios})  and
$F_m$, the measured ones.
```

**p. 11 l. 10: "test" -> "tests";**

Changed

**p. 13 Fig. 7: I can't understand to what the x-axis refers. Brewers #157 and #183 are not listed in Table 1;**

Correct

We have added an explanation in the figure caption. Furthermore, both figures have been merged into a single one

[Figure]

```
    \caption{Box plot of the ozone percentage deviation from the mean of
the RBCC-E triad reference Brewer#157,Brewer#183 and Brewer#185 before
(left panel)  and after (right panel) the X RBCC-E campaign at El
Arenosillo in 2015, grouped by ozone slant columns ranges. The color
indicates the intervals used for the averaging of the observations- blue,
lower than  400 DU; red, between 400 and 700 DU; green, between 700 and
1000 DU; pink, between 1000 and 1200 DU; and purple for OSC > 1200 DU.
```

**p. 15 both figure caption and l. 7: the text was not updated to clarify my previous question why Brewer #151 cannot be considered an "operative instrument";**

*We consider that an operative instrument is one that is capable of sending reliable data routinely to eubrewnet or the woudc. This is not the case of Brewer #151, because of the huge change in the SL record (note that more than 300 R6 units are outside the limits of the axes). This  indicates that Brewer #151 cannot be considered operative.*

*Calibration report:*
https://docs.google.com/document/d/1tD-acUKRj-25Tyuv0DAW1QwSqEyqt2-fuXqAY8EeVAI/edit?usp=sharing

We have include this paragraph in the article:

In the following analysis we do not consider non-operative instruments, an operative instrument is one which is capable of providing reliable data during the campaign suitable for submission to databases like EUBREWNET or WOUDC (Word Ozone and UV Data Center). This is not the case for \#151, because the instrument  shows a huge change on the SL record  from the last calibration (300 R6 units while the instrument range is +/- 60 DU), indicating serious instrumental issues.

**p. 15 l. 21: "Sec" -> "Sect";**

Changed

**p. 16 l. 12-27: why is a new topic (two parameters vs 1 parameter method) introduced in the conclusions?? I would suggest the authors to simply summarise the main topic of the paper, mentioning some quantitative results, instead of introducing new concepts. (Anyway, l. 21: "base" -> "based")**

We moved the topic to the previous section. The conclusion for the calibration is that all the Brewer instruments can be calibrated with one point calibration.

**p. 22 Fig. 11: the x-axis label of the upper panel is missing and the axis is not the same as the one below;**

corrected

**p. 22 caption of Fig. 11: "campaing" -> "campaign";**

corrected

**p. 25 table 2: Brewer #145 is missing in Table 1, therefore you must explain which Brewer is.**

Corrected the caption of Table 2.

Reference Comparison during RBCC-E campaigns, Brewer #017 is the travelling reference from International Ozone Service (IOS), Brewer #158 is the travelling reference from Kipp Zonen, and Brewer #145 from Environmental Canada is a double Brewer directly calibrated using as reference the World Reference Triad which participated in the previous RBCC-E campaigns.

**p. 25 table 2: Brewer #145 is missing in Table 1, therefore you must explain which Brewer is.**

Corrected the caption of Table 2.

Reference Comparison during RBCC-E campaigns, Brewer #017 is the travelling

[revised manuscript text omitted]